# A calcineurin-mediated scaling mechanism that controls a K⁺-leak channel to regulate morphogen and growth factor transcription

Chao Yi[1,2†], Tim WGM Spitters[1‡], Ezz Al-Din Ahmed Al-Far[3‡], Sen Wang[1,2], TianLong Xiong[1,2], Simian Cai[1], Xin Yan[1], Kaomei Guan[3], Michael Wagner[3,4], Ali El-Armouche[3], Christopher L Antos[1,3*]

[1]School of Life Sciences and Technology, ShanghaiTech University, Shanghai, China; [2]CAS Center for Excellence in Molecular Cell Science, Shanghai Institute of Biochemistry and Cell Biology, Chinese Academy of Sciences, University of Chinese Academy of Sciences, Shanghai, China; [3]Institut für Pharmakologie und Toxikologie, Technische Universität Dresden, Dresden, Germany; [4]Klinik für Innere Medizin und Kardiologie, Herzzentrum Dresden, Technische Universität Dresden, Dresden, Germany

*For correspondence:
clantos@shanghaitech.edu.cn

†These authors contributed equally to this work
‡These authors contributed equally to this work

Competing interests: The authors declare that no competing interests exist.

**Abstract** The increase in activity of the two-pore potassium-leak channel Kcnk5b maintains allometric juvenile growth of adult zebrafish appendages. However, it remains unknown how this channel maintains allometric growth and how its bioelectric activity is regulated to scale these anatomical structures. We show the activation of Kcnk5b is sufficient to activate several genes that are part of important development programs. We provide in vivo transplantation evidence that the activation of gene transcription is cell autonomous. We also show that Kcnk5b will induce the expression of different subsets of the tested developmental genes in different cultured mammalian cell lines, which may explain how one electrophysiological stimulus can coordinately regulate the allometric growth of diverse populations of cells in the fin that use different developmental signals. We also provide evidence that the post-translational modification of serine 345 in Kcnk5b by calcineurin regulates channel activity to scale the fin. Thus, we show how an endogenous bioelectric mechanism can be regulated to promote coordinated developmental signaling to generate and scale a vertebrate appendage.

## Introduction

Tissue scaling involves the coordinated control of developmental programs, since anatomical structures consist of different tissues that form in a coordinated manner and grow proportionally with each other and with the body. While there are several developmental signals known to regulate cell proliferation and tissue formation, mechanisms that concomitantly activate several developmental signals to synchronize the growth of multi-tissue appendages and organs in a manner that is coordinated with body proportions remain poorly defined.

There is growing evidence that several biological phenomena involved in tissue generation and growth are influenced by electrophysiological changes in 'non-excitable' cells (*Sundelacruz et al., 2009*). Several cell behaviors are affected by the addition of electric currents (*McCaig et al., 2005*): cell migration, cell proliferation, cell differentiation, gene transcription and consequently tissue formation are all altered by the application of an exogenous current (*Baer and Colello, 2016*; *Bartel et al., 1989*; *Blackiston et al., 2009*; *Borgens et al., 1977*; *Geremia et al., 2007*;

**eLife digest** Organs, limbs, fins and tails are made of multiple tissues whose growth is controlled by specific signals and genetic programmes. All these different cell populations must work together during development or regeneration to form a complete structure that is the right size in relation to the rest of the body. Growing evidence suggests that this synchronicity might be down to electric signals, which are created by movements of charged particles in and out of cells.

In particular, previous work has identified two factors that control the development of fins in fish: the Kcnk5b potassium-leak channel, which allows positive ions to cross the cell membrane; and an enzyme called calcineurin, which can modify the activity of proteins. Kcnk5b and calcineurin seem to play similar roles in the proportional growth of the fins in relation to the body, but exactly how was unknown.

To investigate this question, Yi et al. used genetically modified zebrafish to show how the Kcnk5b channel could control genes responsible for appendage growth. However, their tests on different cell types revealed that potassium movement through the Kcnk5b channel leads to different sets of developmental genes being turned on, depending on the tissue type of the cell. This could explain how one type of signal (in this case, movement of ions) can coordinate the growth of a wide range of tissues that use different combinations of developmental genes to form. Kcnk5b therefore appears to coordinate the regulation of the various combinations of genes needed for different fin tissues to develop, so that every component grows in a proportional, synchronized manner. Yi et al. also showed that calcineurin can modify the Kcnk5b channel to control its activity. In turn, this affects the movement of potassium ions across the membrane, changing electrical activity and, as a consequence, the proportional growth of the fin.

Further work should explore how Kcnk5b and calcineurin link to other signals that regulate the size of fins and limbs. Ultimately, a finer understanding of the molecules controlling the growth of body parts will be useful in fields such as regenerative medicine or stem cell biology, which attempt to build organs for clinical therapies.

*Sundelacruz et al., 2009*; *Yasuda, 1974*; *Zhao et al., 2002*). The culmination of these findings have led to the hypothesis that bioelectrical fields exist that have higher order organizational non-cell-autonomous properties in the development of anatomical structures (for review, see *Levin, 2014*; *Messerli and Graham, 2011*].

As a regulator of membrane potential, $K^+$ conductance is an important component of the electro-physiological properties of cells. Evidence that illustrates the importance of $K^+$ conductance in tissue formation comes from studies in which disruption of inward rectifying $K^+$ channels of the Kir2 family can cause cranial facial defects, abnormal number of digits and reduced digit size (*Andersen et al., 1971*; *Canún et al., 1999*; *Sansone et al., 1997*; *Tawil et al., 1994*; *Yoon et al., 2006a*; *Yoon et al., 2006b*; *Zaritsky et al., 2000*). A striking finding concerning the coordinated control of cell behavior is the formation of eye structures by overexpressing different ion channels that alter membrane potential in early *Xenopus* embryos (*Pai et al., 2012*): overexpression and activation of a glycine-gated chloride channel in cells that form the eye interferes with eye formation, while overexpression of a dominant-negative $K^+$-ATP channel simulates ectopic eye formation even in unexpected locations on the body (*Pai et al., 2012*). These findings illustrate that changes in the membrane potential of cells can have significant impacts on the development of anatomical structures. However, how electrophysiological information associates with the multiple necessary signals that control formation and/or growth of multi-tissue structures remains unclear.

The development of body structures not only involves forming tissues, it also involves coordinating the growth of each contributing tissue cell. To form organs that correctly scale with the body, each tissue grows either isometrically (grows with the same rate as the body) or allometrically (disproportionally grows in relation to the growth of the body). The zebrafish mutants *another long fin* (*alf*), *long fin* (*lof*), and *schleier* (*schl*) display continued allometric growth of each appendage from the juvenile stage into the adult stage (*Lanni et al., 2019*; *Perathoner et al., 2014*; *Stewart et al., 2020*). The dominant allometric growth phenotype of *alf* is due to mutations in the transmembrane pore region of *kcnk5b* (*Perathoner et al., 2014*), encoding a two-pore $K^+$-leak channel that

regulates membrane potential by outward flow of K⁺ from the cell (*Goldstein et al., 2001*). The dominant phenotype of *lof* is linked to elevated expression of a voltage-gated potassium channel *Kcnh2a* (*Stewart et al., 2020*). The phenotype of *schl* is due to dominant-negative mutations in *kcc4* (*Lanni et al., 2019*), encoding a K⁺-Cl⁻ cotransporter that regulates intracellular K⁺ levels in a chloride-dependent manner (*Marcoux et al., 2017*). *alf*, *lof*, and *schl* demonstrate the importance of K⁺ conductance and ultimately of electrophysiological signals for the correct body-to-appendage proportions. Despite the connection between K⁺ conductance and the proportional growth of the fins, it remains unclear how K⁺-mediated signal translates into coordinated growth of the fish appendage and how any K⁺ channel is regulated to scale tissue.

We show that activity of the single two-pore K⁺-leak channel Kcnk5b is sufficient to induce the activation of at least two components, Shh and Lef1, of two important morphogen pathways, not only in the adult fin but also in the larva. Our data also indicates that this induction is cell autonomous, arguing that increases in membrane potential caused by Kcnk5b regulate growth through intracellular regulation of these developmental pathways. Overexpression of *kcnk5b* or one of two other two-pore *kcnk* channels in different mammalian cell lines showed variable activation of genes belonging to different developmental signal transduction mechanisms, supporting the conclusions that the different developmental mechanisms needed to scale fish appendages can be regulated by the same electrophysiological change induced by potassium leak and that the combinatorial activation of the different developmental mechanisms by Kcnk5b is cell-type dependent. Lastly, we provide evidence for how post-translational modification of Kcnk5b at Serine345 by calcineurin regulates its electrophysiological activity and consequently the scaling of zebrafish fins. Thus, we describe an endogenous cell-autonomous mechanism through which electrophysiological signals can induce and coordinate specific morphogen and growth factor signals to mediate the scaling of an anatomical structure.

## Results

### Kcnk5b is sufficient to induce several developmental gene programs in different cell types in adult and larva

Mutations in the two-pore K⁺-leak channel Kcnk5b that increase its activity lead to enhanced growth of the zebrafish appendages (*Perathoner et al., 2014*). While this finding implicates the importance of bioelectric signaling in appendage scaling, it remains unknown how the activity of a single K⁺ channel is integrated with the developmental controls that generate new appendage tissues. Growth of any appendage involves the coordinated activation of specific morphogen and growth factor pathways: Shh, β-catenin-dependent Wnt, Bmp, Fgf, and Retinoic acid. Therefore, to begin to determine how this channel is involved in the coordinated growth of the entire fin, we generated transgenic zebrafish that expresses *kcnk5b* under the control of a conditionally inducible promoter (heat-shock promoter) to temporally activate this channel in adult fins. After a single 10-min heat-shock pulse of the Tg[*hsp70:kcnk5b*-GFP] transgene, we observed significant activation of *shh* and *lef1* (β-catenin-dependent Wnt) (*Figure 1Aa*; *Figure 1—figure supplement 1A*), as well as an increase in *aldh1a2* (retinoic acid) (*Figure 1Ab*) within 6 hr by qRT-PCR, while *pea3* (Fgf) was slightly increased and *msxb* (BMP) was not significantly changed (*Figure 1A*). All genes returned to control levels by 12 hr after the single pulse (*Figure 1B*). The induction of the developmental genes coincided with the temporal expression of the *kcnk5b*-GFP transgene (*Figure 1C*), whose expression emerged as a lattice pattern (*Figure 1—figure supplement 1B–D*), indicating that Kcnk5b-GFP was localized and functioned at cell membranes. When we maintained chronic expression of the transgene by heat shocking the caudal fin for 10 min once per day for 3 days, we observed expression of *lef1*, *shh*, *aldh1a2* as well as *pea3* and *msxb* over controls (*Figure 1D*), which included Shh's patched receptors and slight up-regulation of *bmp2b* (*Figure 1—figure supplement 1E*). Together, these data show that Kcnk5b is sufficient to induce the transcription of certain developmental genes as though it were a part of their signaling mechanisms. Furthermore, the initial activation of *shh*, *lef1*, and *aldh2a* (*Figure 1A*) followed by later upregulation of *pea3* and *msxb* after continued *kcnk5b* overexpression (*Figure 1C*) suggest that there is a hierarchical activation of developmental mechanisms that will ultimately lead to the complete allometric growth program.

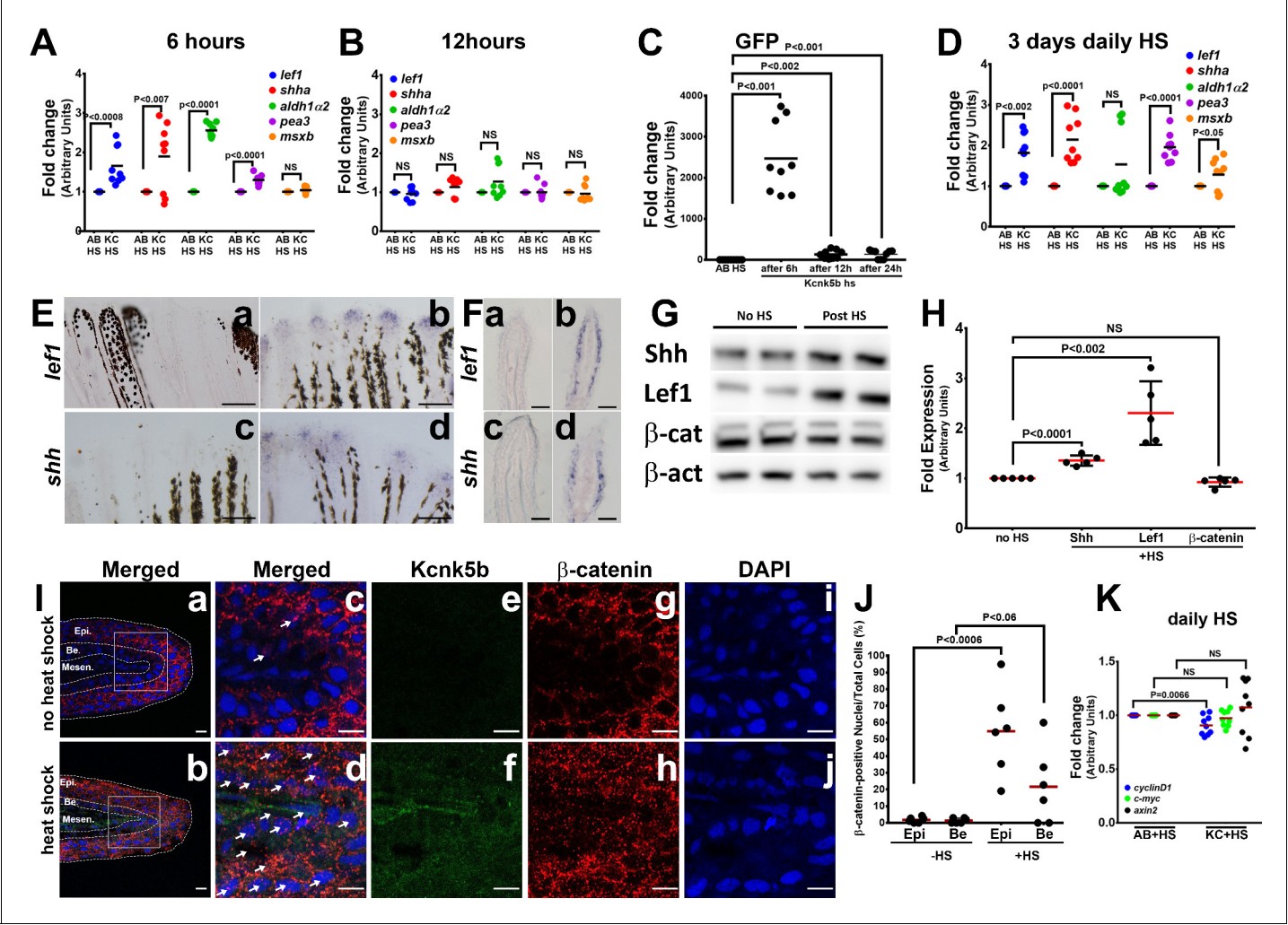

**Figure 1.** Kcnk5b induces a partial developmental gene response in uninjured adult fins. (**A**) qRT-PCR results of *shh* and *lef1, aldh1α2, pea3,* and *msxb* expression from caudal fins of 6-month-old wild-type (AB) and Tg[*hsp70:kcnk5b*-GFP] zebrafish 6 hr after heat shock (HS)of the caudal fins. (**B**) qRT-PCR results of *shh* and *lef1, aldh1α2, pea3* and *msxb* expression from caudal fins of 6-month-old wild-type (AB) and Tg[*hsp70:kcnk5b*-GFP] zebrafish 12 hr after heat-shock induction of the the caudal fins. (**C**) qRT-PCR results for GFP from the transgenic fish line Tg[*hsp70:kcnk5b*-GFP] at the indicated time points relative to the single heat-shock pulse of AB wild-type and transgenic fish. AB HS for each time point were always set at onefold, so they are represented as only one group in the graph. (**D**) qRT-PCR results for several genes in the caudal fin from daily heat-shock pulse of AB wild-type and Tg [*hsp70:knck5b*-GFP] over 3 days. (**E**) In situ hybridization experiments on fins show expression of *shh* (**a,b**) and *lef1* (**c,d**) in heat-shocked non-transgenic control fish (**a,c**) and heat-shocked Tg[*hsp70:kcnk5b*-GFP](**b,d**). (**F**) Cross-sections through fin rays show expression of *lef1*(**a,b**) and *shh* (**c,d**) before (**a,c**) and after (**b,d**) heat-shock induction of Tg[*hsp70:kcnk5b*-GFP]. (**G**) Representative images of Western blots show expression of Shh, Lef1 and β-catenin before and 3 days after 10 min daily heat-shock induction of *kcnk5b*-GFP in the fin. (**H**) Graphed measurements results of Western blots. (**I**) Confocal planes (0.45 µm) of immunohistochemistry stained 10 µm sections of control (no heat shock) or transgene-induced (heat shock) uninjured fins for merged GFP from Tg[*hsp70:kcnk5b*-GFP] and β-catenin (red) (**a–d**: Merged), GFP from Tg[*hsp70:kcnk5b*-GFP] (**e,f**: green) β-catenin (**g,h**: red) and DAPI (**i,j**: blue)of fin cross-sections of transgenic Tg[*hsp70:kcnk5b*-GFP] animals without heat shock (**a,c,e,g,i**) or after heat shock (**b,d,f,h,j**). White boxes in (**a** and **b**) show location of magnified panels of **c,e,g** and **d,f,j**, respectively. Overlapping DAPI and β-catenin staining indicated by white arrows. Epi.' refers to the outer multilayered epidermis, 'Be.' as the underlying basal epithelial layer, and 'Mesen.' refers to the underlying mesenchymal tissues. (**J**) Graphed measurements of DAPI stained nuclei containing staining of β-catenin. (**K**) qRT-PCR results for the indicated genes in the caudal fin from AB non-transgenic and Tg[*hsp70:knck5b*-GFP] fish after daily heat-shock pulses over 3 days. Scale bars are 50 µm (**D**), 1 mm (**E**), 10 µm (**H**). The data for each experiment represent three or more separate experiments. The data points show all technical replicates. Student's T-test used for the tests of significance between indicated experimental groups.

The online version of this article includes the following figure supplement(s) for figure 1:

**Figure supplement 1.** Induction of developmental genes after induction of *kcnk5b* expression.

**Figure supplement 2.** Protein expression of selected developmental genes and β-catenin subcellular distribution.

To examine the spatial expression of the two genes most responsive to Kcnk5b, we performed in situ hybridization experiments for *shh* and *lef1*. Compared to control fins (Figure 3Ea, c), localization of *shh* and *lef1* was in the distal tip of the fin (*Figure 1Eb, d*) where growth normally occurs, and cross-sections through the fins showed that Kcnk5b-mediated induction of these genes occured in the epidermal/epithelial tissues (*Figure 1Fb*,d). We also assessed increases in Shh and Lef1 protein levels after the heat-shock induction of *kcnk5b*-GFP (*Figure 1G,H*; *Figure 1—figure supplement 2A,B*). Because Lef1 conveys β-catenin-dependent Wnt signaling by acting as a transcriptional plat-form for β-catenin, we examined protein expression of β-catenin and observed no significant differences in its overall levels (*Figure 1G,H*; *Figure 1—figure supplement 2C*). However, when we examined β-catenin protein distribution in the fin tissues by immunohistochemistry staining, we observed an increased number of nuclei with β-catenin co-staining in the outer epidermis layers and basal epithelium of the fin (*Figure 1I,J*; *Figure 1—figure supplement 2E–J*) despite no significant differences in the measured β-catenin-associated fluorescence intensities (*Figure 1—figure supplement 2K*). To test whether the redistribution of β-catenin leads to the activation of known β-catenin-dependent genes, we performed qRT-PCR of *axin2*, *cyclin D*, and *c-myc*. We observed that none of these candidates were activated (*Figure 1K*), which indicates that while Lef1 expression increases, β-catenin-dependent Wnt signaling is not directly activated by Kcnk5b. Together, all the expression results indicate that Kcnk5b activity promotes the transcription of specific components of a limited number of developmental pathways in the adult fin.

To test whether increasing the activity of Kcnk5b has the same transcriptional effect on these developmental pathways in another in vivo context, we induced the expression of *kcnk5b* in the zebrafish larva (*Figure 2—figure supplement 1A*). We observed that *lef1*, *shh*, *adlh2a*, *pea3*, *and msxb* expression levels were increased by induction of *kcnk5b* compared to heat-shocked non-trans-genic control fish (*Figure 2A*). To further explore the effect of Kcnk5b activity on *shh* transcription in developing fins, we performed in situ hybridization experiments on fish embryos and examined *shh* expression in the developing pectoral fin buds. Compared to the emerging expression of *shh* in the fin buds of heat-shocked non-transgenic embryos (*Figure 2Ba*,c), we observed *kcnk5b* expression induced increases in the rate of *shh* mRNA detection (*Figure 2Bb*,d) and in the area of *shh* expression (*Figure 2Be-h*). Thus, Kcnk5b activity appears to increase both the intensity (*Figure 2Bi*) and the range of *shh* expression (*Figure 2Bj*). We further assessed Lef-1-dependent transcription by crossing the heat-shock-inducible transgenic Tg[*hsp70:knck5b*-GFP] line with the transgenic Lef1 reporter line Tg[*7XTCF-Xla.sam*:mCherry]. While the double-transgenic fish Tg[*hsp70:kcnk5b*-GFP; *7XTCF-Xla.sam*:mCherry] displayed limited expression of mCherry before heat-shock induction of the *kcnk5b*-GFP transgene (*Figure 2Ca*,b,g), after heat shock, double-transgenic Tg[*hsp70:kcnk5b*-GFP; *7XTCF-Xla.sam*:mCherry] fish showed a broad increase in reporter mCherry expression over single-transgenic Tg[*7XTCF-Xla.sam*:mCherry] (*Figure 2Cd*,e,g). From histological cross-sections of double-transgenic larva, we observed that mCherry was upregulated broadly in the body of the ani-mal (*Figure 2D*). Furthermore, we observed an increase in the length of the caudal finfold along the anterioposterior axis of 5 dpf larva after induction of *kcnk5b*-GFP by single daily 10 min pulses over 3 days (*Figure 2F*; *Figure 2—figure supplement 1C*), while the caudal finfold lengths along the dor-soventral axis of appeared not change (*Figure 2G*). However, we observed that when we compared both anterioposterior and dorsoventral axes measurements in relation to body length, we observed that in both instances the finfold dimensions proportionally increased (*Figure 2H,I*), which was in part associated with decreases in the lengths of the bodies (*Figure 2J*). These results showed that in relation to the body, Kcnk5b activity promoted proportional increases in the finfold dimensions. Our observation that the activity of this channel decreased the length of the body, suggests that the growth-promoting effects of Kcnk5b activity may be limited to the finfolds and appendages. Together, these results indicate that Kcnk5b activity is sufficient to promote the transcription of specific developmental genes in several different tissue types to promote allometric growth of the finfold.

## Activation of the Lef1-dependent transcription by Kcnk5b is cell autonomous in several different tissues

Previous work implicates bioelectric intercellular communication as a mechanism for how bioelectricity can influence tissue growth (*McLaughlin and Levin, 2018*), and changes in K$^+$ channel activity have been shown to regulate different cell behaviors in a non-cell-autonomous manner

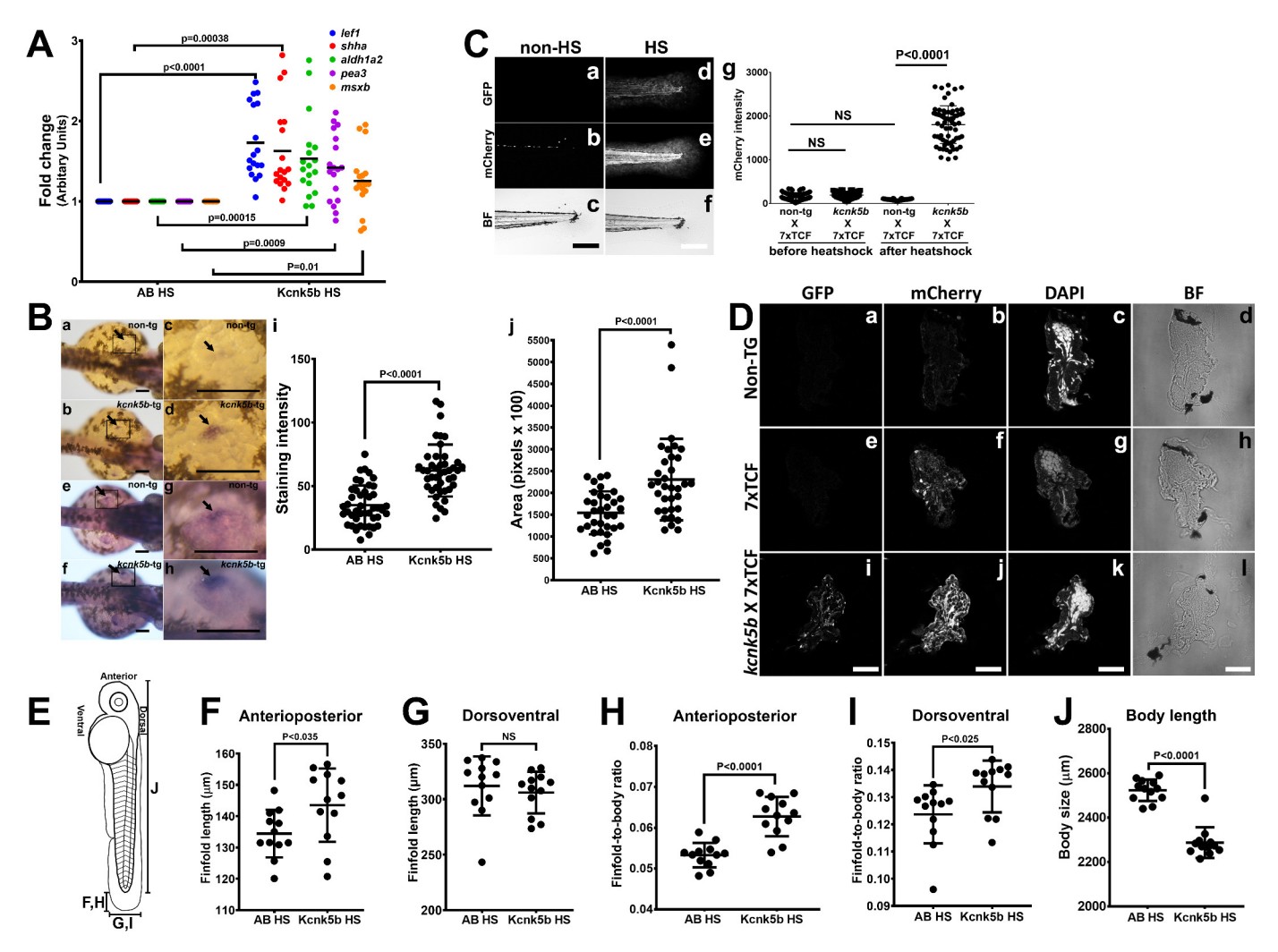

**Figure 2.** Kcnk5b induces *shh* and *lef1* in zebrafish larva. (**Aa,b**) Comparison of gene expression from mRNA isolated from complete three dpf zebrafish larva harboring Tg[*hsp70:knck5b*-GFP] 6 hr after heat-shock (HS) induction. (**B**) In situ hybridization experiments on 2 dpf embyros for *shh* start to detect expression in the cells of the zone of polarity in the early pectoral fin bud by 60 min of incubation of the final staining reaction (**a–d**) or after overnight staining (**e–h**). Measurements of *shh* expression levels by computer-program-based detection of pixel level intensities (**i**) or by the stained area (**j**) in the early pectoral fin bud. Panels c,d,g,h are enlarged regions (open black boxes) in panels a,b,e,f. Arrows indicate *shh* expression in the developing fin buds. (**C**) Double transgenic 3dpf larva harboring Tg[*hsp70:kcn5b*-GFP] and hemizygous Tg[*7XTCF-Xla.sam*:mCherry] either before heat shock (**a–c**) or 12 hr after heat shock (**d–f**). (**Cg**) Measurements of mCherry intensity levels of non-transgenic (non-tg), *kcnk5b*-transgenic (*kcnk5b*) fish and transgenic fish harboring the Lef-1-dependent Tg[*7XTCF-Xla.sam*:mCherry] reporter (7xTCF) before and 12 hr after heat shock. (**D**) Cross sections through the trunks of non-transgenic (**a–d**) and Tg[*7XTCF-Xla.sam*:mCherry] (**e–h**) and Tg[*hsp70:kcn5b*-GFP]; Tg[*7XTCF-Xla.sam*:mCherry] double-transgenic (**i–l**) fish lines after heat shock. (**D**) Measurements from dorsal to ventral of the caudal finfold of Tg[*7XTCF-Xla.sam*:mCherry] and Tg[7XTCF-Xla.sam:mCherry] X Tg[*hsp70:kcnk5b*-GFP] sibling larva after heat shock. (**E**) Diagram of the measurement axes of the larva. (**F**) Length measurements from anterior-most point of the caudal finfold to its distal-most tip of 5 dpf larva. (**G**) Length measurements of the caudal finfold from the dorsal-most point to the ventral-most tip of 5 dpf larva. (**H**) Finfold anterioposterior length-to-body ratios of the caudal finfold of 5 dpf larva. (**I**) Finfold dorsoventral length-to-body ratios of the body ratios. (**J**) Body length measurements of 5 dpf larva as diagramed in (**E**). Scale bars are 100 µm (**B,C**) and 20 µm (**D**). The data for each panel represent three or more experiments. The data points show all technical replicates. Student's T-test used for all test of significance between the indicated experimental groups.

The online version of this article includes the following figure supplement(s) for figure 2:

**Figure supplement 1.** Controls for shh expression and transgenic expression of Kcnk5b in embryos.

(*Morokuma et al., 2008*; *Pai et al., 2015*). The broad activation of the Lef1-dependent Wnt reporter in several tissues (*Figure 2*) and Kcnk5b's ability to scale all the tissues of the fin appendages suggest that Kcnk5b acts via non-cell autonomous communication among cells. To determine whether the observed Kcnk5b-mediated induction of gene expression is due to intercellular communication (e.g. through extracellular ligands such as Wnt) or due to cell autonomous activation of transcription, we transplanted cells from Tg[*hsp70:kcnk5b*-GFP; *7XTCF-Xla.sam*:mCherry] transgenic embryos into embryos harboring only the Tg[*7XTCF-Xla.sam*:mCherry] transgene and then raised mosaic embryos and larva (*Figure 3A*). Analyses of the mosaic larva showed the previously reported developmental expression of the Lef1-dependent reporter before heat shock (*Figure 3Ba,e,i,m*) (*Moro et al., 2012*). However, after heat-shock induction of the Tg[*hsp70:kcnk5b*-GFP] transgene (*Figure 3Bb,f,j, n*), we observed ectopic activation of *7XTCF-Xla.sam*:mCherry reporter only in donor cells of chimeric 48 hpf and 72 hpf fish (recipient *7XTCF-Xla.sam*:mCherry fish harboring transplanted cells from Tg[*hsp70:kcnk5b*-GFP;*7XTCF-Xla.sam*:mCherry] embryos). GFP-mCherry-positive cells appeared in tissues in the head (*Figure 3Bb*,c,d), in skeleton surrounding the eye (*Figure 3Bf*,g,h), in trunk muscles (*Figure 3Bj*,k,l) and in skin (*Figure 3Bn*,o,p). From closer inspection, we observed co-expression in neurons in the head (*Figure 3C*,a-c), in the ectodermal bones of the skull (*Figure 3C*, d-f), mandible bone and cartilage (*Figure 3C*, g-i), mesenchyme surrounding the otic vesicle (*Figure 3C*, j-l), epithelial cells in the finfold (*Figure 3C*, o-r) and individual striated muscle cells of the trunk (*Figure 3C*, r-t). We counted the number of GFP and mCherry positive cells in the different tissues and observed that all Kcnk5b-GFP-positive cells were mCherry positive (*Figure 3C*). Moreover, in all tissues, the ectopic mCherry expression was always limited to the Kcnk5b-positive cells (*Figure 3B–E*; *Figure 3—figure supplement 1G–J*). Together, these data support two conclusions: one, the activation of the Lef1-dependent reporter by Kcnk5b is cell autonomous; and two, Kcnk5b is able to promote the expression of the Lef1 reporter in diverse tissue types.

As a K$^+$-leak channel, Kcnk5b's activity should decrease intracellular K$^+$ levels. We performed Fluorescence Lifetime Microscopy (FLIM) analysis with an established genetic sensor for K$^+$ to measure intracellular K$^+$ levels (*Shen et al., 2019*). This sensor uses the FRET potential between two fluorophores that are joined by a K$^+$-binding linker. Changes in FRET due to K$^+$ binding results in changes in the fluorescence lifetime of the fluorophores, which allows for the assessment of intracellular K$^+$ levels. Transfection of the channel in Human Embryonic Kidney HEK293T cells (*Figure 4—figure supplement 1A–L*) resulted in significant increase in CFP fluorescence lifetime due to decreased FRET of the sensor compared to control transfected cells (*Figure 4A*, a-c,g), which indicated reduced intracellular K$^+$ levels in the cells that express Kcnk5b (*Figure 4A*, d-g). Additional higher resolution assessments along the lateral borders of cells showed similar increases in CFP fluorescence lifetime along the plasma membrane, indicating expected reduction of K$^+$ levels at the cell membrane by active Kcnk5b (*Figure 4—figure supplement 1M–O*).

To test whether activity *kcnk5b* promotes the gene expression profile in mammalian cells that we observed in the zebrafish, we expressed *kcnk5b* either by establishing stable HEK293T (HEK) cells lines that either express GFP or zebrafish *kcnk5b*-GFP or by transient transfections. From qRT-PCR analyses comparing HEK cells expressing either GFP or *kcnk5b*-GFP, we observed an increase in SHH and PEA3 expression (*Figure 4Ba*; *Figure 4—figure supplement 2A*) and the down-regulation of LEF1, ALDH1a2 and MSX1 (*Figure 4Bb*; *Figure 4—figure supplement 2B*). To determine whether this transcriptional response is specific to Kcnk5b or is a general response to two-pore K$^+$-leak channels, we transfected cells with one of two K$^+$-leak channels Kcnk9 and Kcnk10 (*Figure 4—figure supplement 2C,D*). Transfection of HEK cells with these two other channels resulted in a similar HEK-cell transcriptional profile as Kcnk5b for SHH and FGF (*Figure 4C*), indicating that this transcriptional response to Kcnk5b is a response to the electrophysiological changes associated with intracellular K$^+$ leak.

The differences between the transcriptional responses of the zebrafish adult, larva, and HEK cells indicate that different cell types will have different responses to Kcnk5b electrophysiological activity. Therefore, we examined the transcriptional responses to Kcnk5b in other mammalian cell lines. In HeLa cells, Kcnk5b induced PEA3 and LEF (*Figure 4D*). In the N2A (neural carcinoma) cell line, we observed the increase of ALDH1a2 (*Figure 4E*) but decreases in SHH, LEF1 and PEA3 (*Figure 4F*). In the MCF7 epithelial carcinoma cell line, Kcnk5b induced ALDH1a2, PEA3, and MSX1 (*Figure 4G*).

When we tested whether Kcnk9 and Kcnk10 produce similar transcription profiles as Kcnk5b in HELA and N2A cells, we observed similar profiles between Kcnk9 and Kcnk10 (*Figure 4H–L*), but

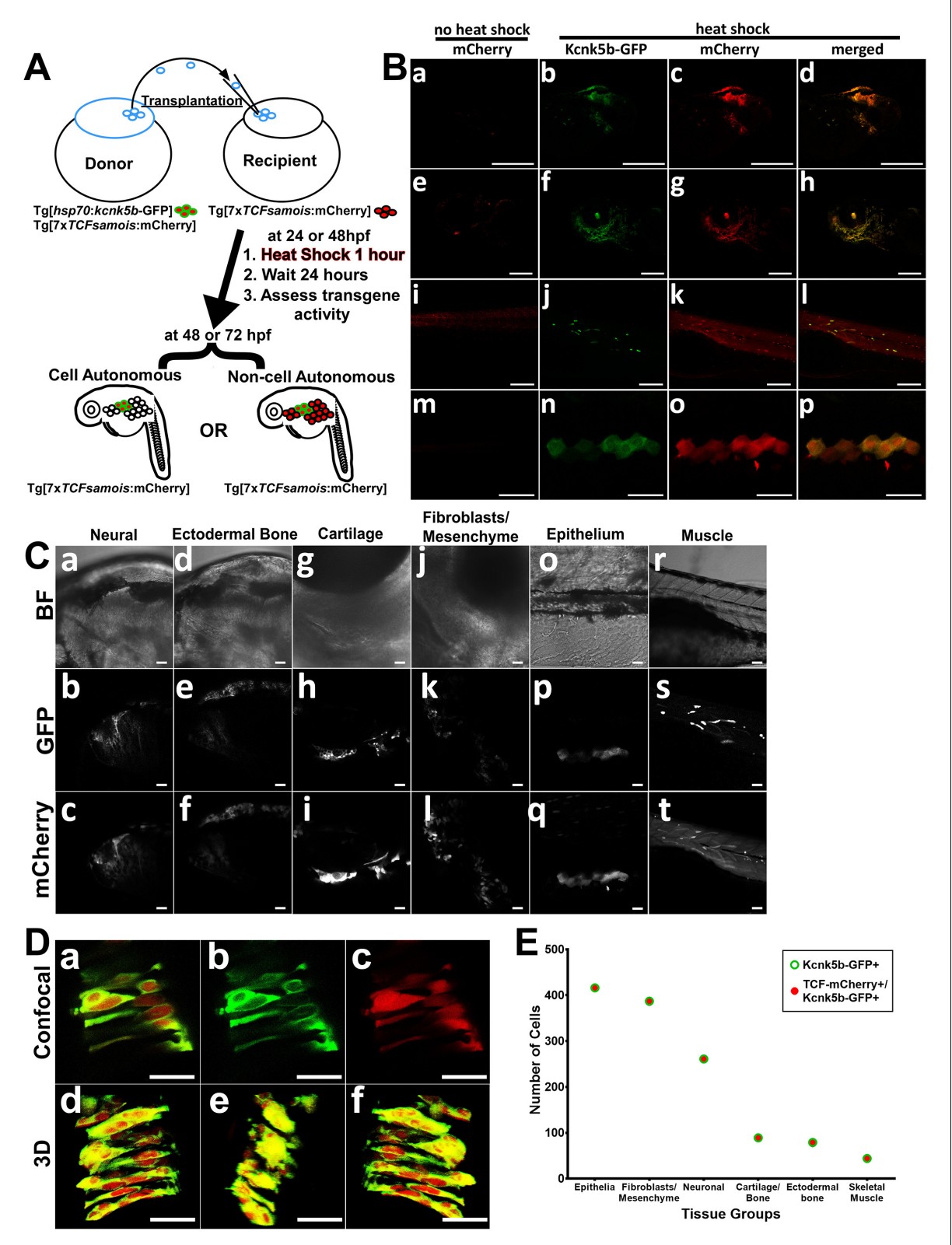

**Figure 3.** Kcnk5b induces Lef-1-dependent transcription in several tissues in a cell autonomous manner. (A) Diagram of transplantation procedure and possible cell-autonomous and non-cell-autonomous outcomes on the expression of the Tg[*7XTCF-Xla.sam:mCherry*] Lef1 reporter after heat-shock induction of the Tg[*hsp70:kcnk5b*-GFP] transgene. (B) Transplantation experiments of donor cells from double transgenic fish harboring Tg[*hsp70: kcnk5b*-GFP] and Tg[7XTCF-Xla.sam:mCherry] into homozygous host embryos harboring only the Tg[7XTCF-Xla.sam:mCherry]. The head (a) eye (e)

*Figure 3 continued on next page*

Figure 3 continued

trunk (i) and finfold (m) of mosaic larva before heat shock induction of *kcnk5b*-GFP expression. Head (**b–d**), eye (**f–h**), trunk (**j–l**), and finfold (**n–p**) of 72 dpf larva at 24 hr after heat shock. (**C**) Bright field images of the head (**a,d**), jaw area (**g**) border tissue of otic vesicle (**j**) and trunk (**o,r**) of 72 hpf larva; GFP expression from Tg[*hsp70:kcnk5b*-GFP](**b,e,h,k,p,s**) and mCherry expression from Tg[*7XTCF-Xla.sam*:mCherry] (**c,f,i,l,q,t**). (**D**) Higher magnification of merged (**a**) and separate GFP (**b**) and mCherry (**c**) channels of cells in the neural tube of 48 hpf embryos. Z-stack composite three-dimensional images before rotation (**d**), rotated 90˚ (**e**) and 180˚ (**f**). (**E**) Total number of positive cells counted in the tissues of all mosaic larva for all mCherry-positive cells from recipient Tg[*7XTCF-Xla.sam*:mCherry] larva, all GFP-positive cells from Tg[*hsp70:kcnk5b*-GFP](open green triangle) and all double positive cells (open blue squares). The data for each experiment represent three or more experiments with two or more biological replicates. Scale bars are equal to 200 μm (Ba-l), 50 μm (Bm-p), 20 μm (C), and 25 μm (D).

The online version of this article includes the following source data and figure supplement(s) for figure 3:

**Source data 1.** Counting Results of cells expressing the mCherry Lef-1 reporter and GFP-labelled Kcnk5b transgene.
**Figure supplement 1.** Expression of Kcnk5b-GFP from Tg[*hsp70:kcnk5b*-GFP] after single heat shock and consequent activation of *7XTCF-Xla.sam*:mCherry.

differences between their outcomes and the transcriptional outcome of Kcnk5b. In HELA cells, the transcriptional response of PEA3 and ALDH2A to Kcnk9 and −10 were different than that of Kcnk5b (*Figure 4D,I–J*). In N2A cells, while we observed similar increases in Aldh2A transcription and similar trends for Pea3 (*Figure 4E,F,K,L*), we also observed the lack of down regulation of Shh, Lef1 and Pea3 (*Figure 4E,F,K,L*). Together, these results reveal that Kcnk channel activity is sufficient to induce the transcription of genes associated with different developmental pathways in different mammalian cells types, and indicate that the downstream consequences of membrane potential changes are not intrinsic to specific signal transduction pathways. We propose that the variability in gene transcription in different cell types may explain why the solitary change in the activity of this channel in all cells of the fin leads to the variable transcriptional responses that promote coordinated growth of a multi-tissue anatomical structure.

## Calcineurin regulates Kcnk5b channel activity and Kcnk5b-mediated gene transcription

We previously showed that the phosphatase calcineurin acts as a molecular switch between isometric and allometric proportional growth of the zebrafish fins (*Kujawski et al., 2014*). The similarities in the phenotypes produced by calcineurin inhibition and by the mutations in Knck5b that enhance Kcnk5b channel activity suggest a direct functional relationship between them (*Kujawski et al., 2014*; *Perathoner et al., 2014*). Based on whole-cell patch-clamp experiments, the mutations in *kcnk5b* that maintain allometric growth of the fins also increased K$^+$ conductance at the plasma membrane (*Perathoner et al., 2014*). Therefore, we hypothesized that calcineurin inhibition will increase in vivo K$^+$ conductance and promote allometric growth of the zebrafish fins (*Figure 5A*). To test whether calcineurin alters the channel activity of Kcnk5b, we examined the activity of Kcnk5b in the presence of calcineurin. Comparison of whole-cell patch-clamp measurements using HEK cells showed that Kcnk5b expression increases current density due to K$^+$ leak from the cells (*Figure 5B*), as we observed from FRET-FLIM intracellular K$^+$ measurements (*Figure 4A*). However, cells co-expressing both *knck5b* and calcineurin decreased the K$^+$ conductance of the cells compared to cells expressing *kcnk5b* alone (*Figure 5B*). We then tested whether inhibition of the endogenous calcineurin activity in the HEK cells by the calcineurin inhibitor FK506 affects Kcnk5b channel activity. We found that FK506 treatment of cells expressing *kcnk5b* resulted in a significant increase in K$^+$ current compared with DMSO-treated *kcnk5b*-expressing cells (*Figure 5C*). These results show that changes in calcineurin activity alter Kcnk5b channel activity in a manner that is constant with the enhanced fin growth induced by calcineurin inhibition and the increased channel activity of the *kcnk5b* zebrafish mutants, which indicates a functional interaction between calcineurin and Kcnk5b.

Calcineurin interacts with its substrates at particular amino acid sequence sites (*Grigoriu et al., 2013*). Our analysis of the amino acid sequence in the C-terminal cytoplasmic tail of Kcnk5b suggests a functional calcineurin binding site (LVIP) is present (*Figure 5A*, red letters). To test for functional interaction at this site, we mutated the amino acid sequence (*Figure 5D*, red letters) and assessed how the mutation affected the ability of calcineurin to regulate the channel. Compared to the decrease in activity of the wild-type channel after co-transfection with calcineurin, co-transfection of the Kcnk5b mutant lacking the calcineurin binding site (Kcnk5bmut + CaN) showed that the

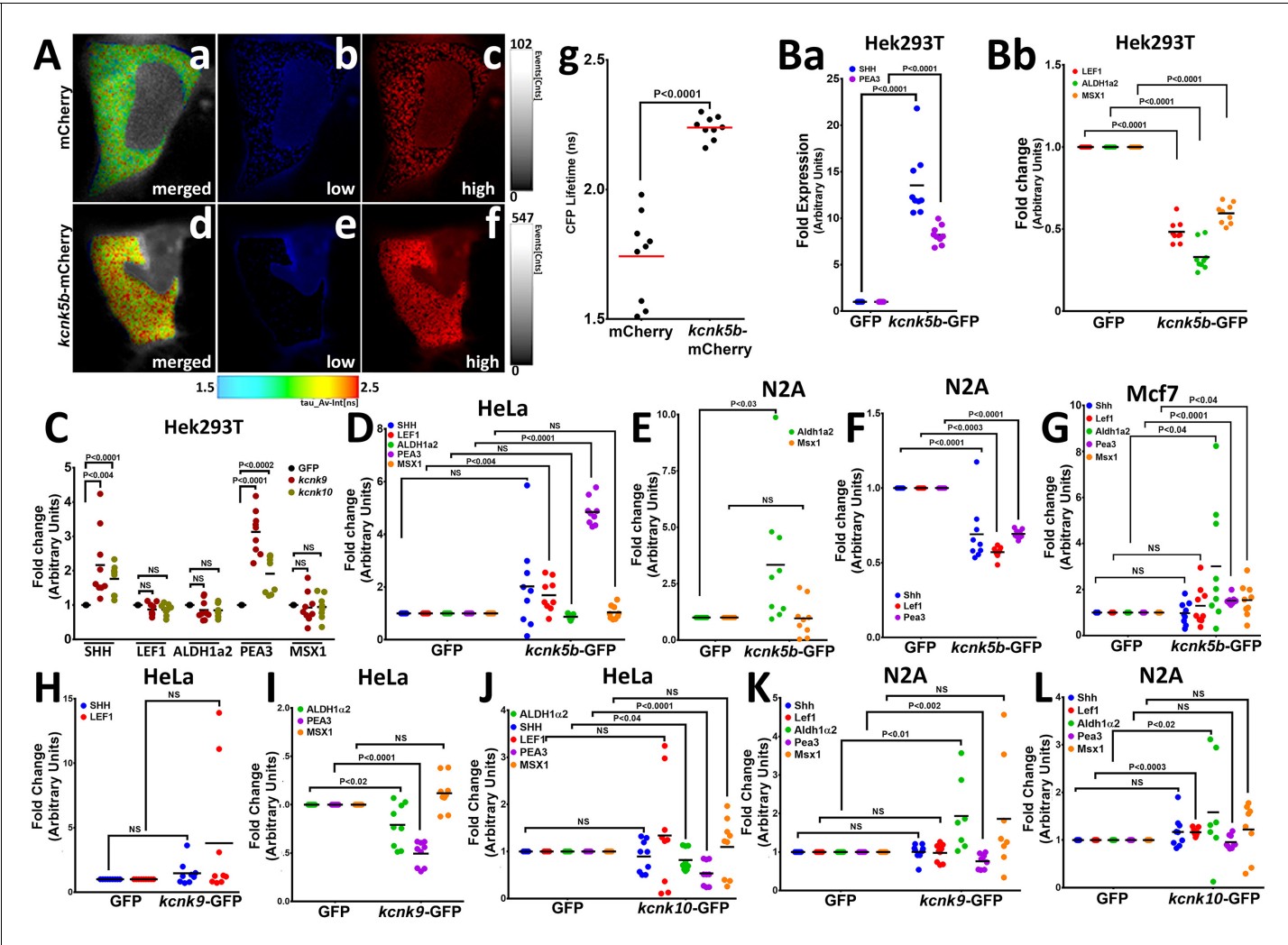

**Figure 4.** Kcnk5b channel activity regulates developmental gene transcription in mammalian cells. (A) FRET-FLIM images after measuring the life time of CFP of the K⁺ FRET reporter KIRIN (**Shen et al., 2019**). The color images indicate the differences in CFP fluorescence lifetime of the K⁺ FRET reporter KIRIN in HEK293T (HEK) cells. Assigned rainbow of colors in the delineated cytoplasm depict the range of numeric values of nanoseconds (ns) of the detected fluorescent lifetime for CFP. Red represents longer lifetime values. Blue represents shorter lifetime values, and the other colors represent intermediary lifetime values. (a) Composite image of all lifetime values of the KIRIN K⁺ reporter in control cells transfected with mCherry. (b) Image of low lifetime values in a control cell. (c) Image of high lifetime values in a control cell. (d) Composite image of all lifetime values of the KIRIN K⁺ reporter in cells expressing *kcnk5b*-mCherry. (e) Image of low lifetime values in cells expressing *kcnk5b*-mCherry. (f) Image of high lifetime values in cells expressing *kcnk5b*-mCherry. (g) Compared to GFP-transfected HEK cells, cells transfected with *kcnk5b*-mCherry show an increase in CFP lifetime due to reduction in intracellular K⁺. (Ba) qRT-PCR for SHH and LEF1 in HEK cells. (Bb) qRT-PCR for ALDH1a2, PEA3 and MSX1 in HEK cells. (C) qRT-PCR for indicated genes in HEK cells expressing GFP, *kcnk9*-GFP or *kcnk10*-GFP 24 hr after transfection. (D) qRT-PCR results in HeLa cells expressing either GFP or *kcnk5b*-GFP 24 hr after transfection. (E,F) qRT-PCR results in N2A cells expressing either GFP or *kcnk5b*-GFP 24 hr after transfection. (G) qRT-PCR results in Mcf7 cells expressing either GFP or *kcnk5b*-GFP 24 hr after transfection. (H,I) qRT-PCR measurement of indicated gene after 24 hr transfection of Kcnk9 in Hela cells. (J) qRT-PCR measurement of indicated gene after 24 hr transfection of Kcnk10 in HeLa cells. (K) qRT-PCR measurement of indicated gene after 24 hr transfection of Kcnk9 in N2A cells. (L) qRT-PCR measurement of indicated gene after 24 hr transfection of Kcnk10 in N2A cells. The data represent three or more experiments, The data points show all technical replicates. Student's T-test was used for tests of significance and the levels of significance are indicated between the experimental groups.

The online version of this article includes the following figure supplement(s) for figure 4:

**Figure supplement 1.** Expression and activity of Kcnk5b-GFP in transfected HEK293T cells.
**Figure supplement 2.** Comparison of Kcnk5b with Knck9 or with Kcnk10.

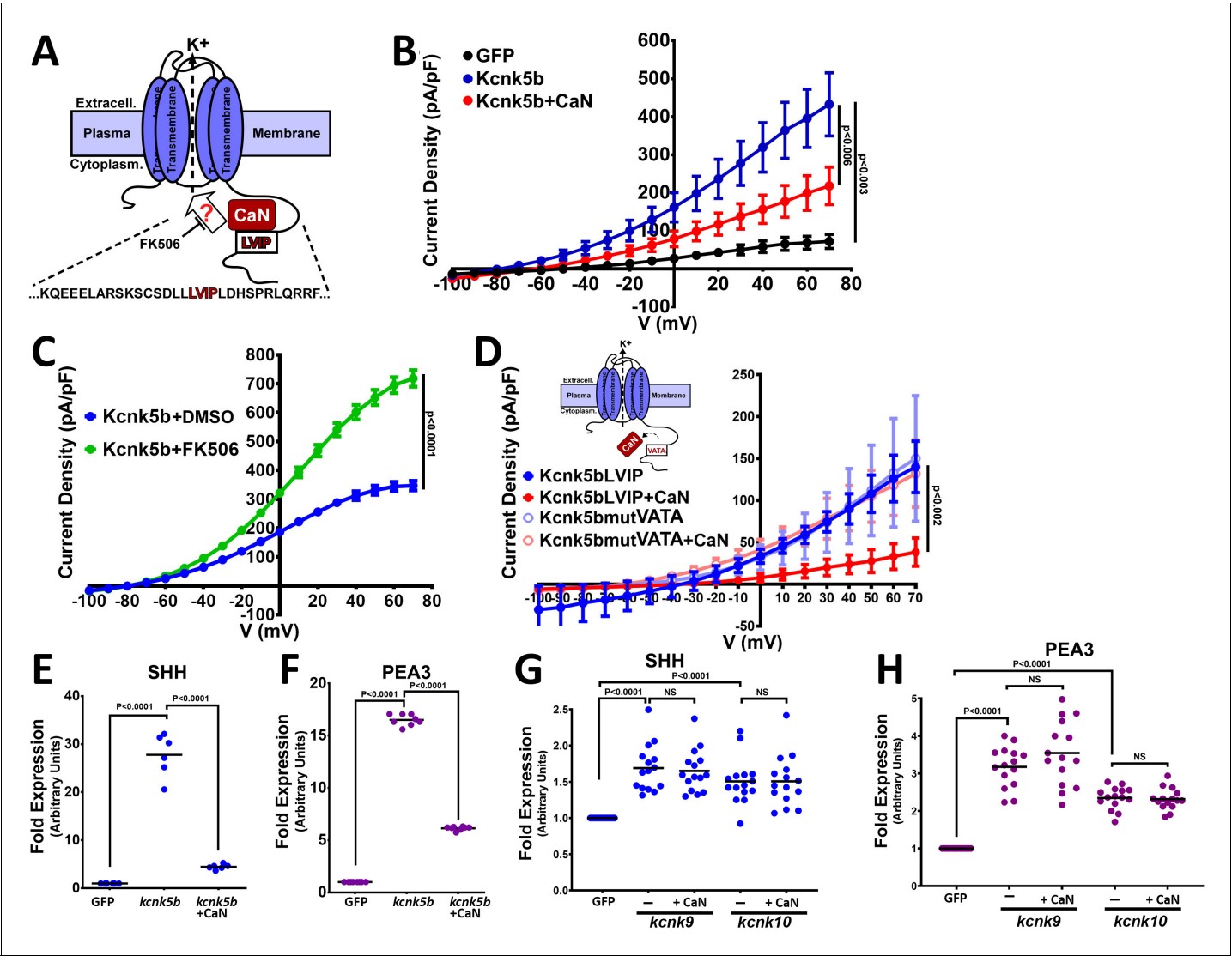

**Figure 5.** Calcineurin functionally interacts and regulates channel activity of Kcnk5b. (**A**) Diagram of hypothetical interaction between Calcineurin (CaN) and Kcnk5b at a consensus calcineurin binding site (LVIP) in Kcnk5b. (**B**) Whole-cell patch clamp of HEK293T (HEK) cells expressing the indicated zebrafish proteins: Calcineurin-mCherry and Kcnk5b-GFP. (**C**) Whole-cell patch-clamp results of cells expressing zebrafish Kcnk5b-GFP and treated either with DMSO or the calcineurin inhibitor FK506. (**D**) Diagram shows mutant Kcnk5b with altered amino acids at putative calcineurin binding site and graph of the Patch-clamp results of the wild-type zebrafish Kcnk5b channel (Kcnk5bLVIP) or mutant Kcnk5b (Kcnk5bmutVATA) lacking the putative calcineurin binding site. Each construct is expressed either with or without calcineurin (CaN). (**E**) qRT-PCR for SHH in HEK cell lines stably expressing either GFP or Kcnk5b-GFP and transfection with calcineurin-mCherry (CaN). (**F**) qRT-PCR for PEA3 in HEK cell lines stably expressing either GFP or Kcnk5b-GFP and transfection with calcineurin-mCherry (CaN). (**G**) qRT-PCR of SHH expression HEK cells after transfection either with GFP, *kcnk9*-GFP or *kcnk10*-GFP with or without calcineurin (CaN). (**H**) qRT-PCR of PEA3 expression HEK cells after transfection either with GFP, *kcnk9*-GFP, or *kcnk10*-GFP with or without calcineurin (CaN). The electrophysiology measurements (**B–D**) are averages with SEM. (**E–G**) Graph panels show averages. The data represent three or more experiments. The averaged data points graphed in D-B have two or more technical replicates per experiment. The data points graphed in E-H show all technical replicates. Student's T test was used to determine the indicated significance (p) values.

The online version of this article includes the following source data for figure 5:

**Source data 1.** Electrophysiological measurement data.

mutation made the channel resistant to calcineurin-mediated inhibition (*Figure 5D*). The resistance of the Kcnk5bmut to calcineurin indicated that the repression of channel activity on Kcnk5b by calcineurin is due to the interaction of these to proteins at the LVIP site.

The regulation of Kcnk5b by calcineurin suggests that changes in calcineurin activity will have an effect on the Kcnk5b-dependent gene expression. To assess whether the activation of SHH by

Knck5b can be altered by calcineurin, we compared the expression of SHH between HEK cells stably expressing GFP and HEK cells stably expressing the Kcnk5b channel as well as between HEK cells stably expressing the channel after transfection with calcineurin. We observed that compared to channel expression alone, co-expression of calcineurin decreased the Kcnk5b-mediated induction of SHH (*Figure 5E*) and PEA3 (*Figure 5F*). To determine whether calcineurin effect on Kcnk5b-medidated SHH expression is specific to Kcnk5b, we transfected HEK cells with Kcnk9 or Kcnk10. Both Kcnk9 and Kcnk10 lack identifiable calcineurin-binding sites (*Figure 4—figure supplement 2C,D*), and we observed that unlike the effect on Kcnk5b, calcineurin had no effect on the induction of SHH (*Figure 5G*) or PEA3 (*Figure 5H*) by Kcnk9 or by Kcnk10, indicating that calcineurin's regulation of the electrophysiological induction of SHH and PEA3 transcription is specific to Kcnk5b.

## Calcineurin regulates Kcnk5b through S345 to scale the fin

As a phosphatase, calcineurin should regulate Kcnk5b by dephosphorylating the channel at specific serine or threonine residues. A specific serine in the C-terminal tail represented a typical consensus serine-proline (Ser345-Pro346) phosphorylation site for calcineurin (*Figure 6A*). Therefore, we hypothesized that calcineurin inhibits the activity the Kcnk5b channel by dephosphorylating this serine. We tested whether rendering the Kcnk5b channel unphosphorylatable at this serine by alanine substitution (S345A) would decrease the channel's activity. Whole-cell patch-clamp experiments of the *kcnk5bS345A* showed a significant decrease in $K^+$ conductance of the channel compared to the wild-type (*kcnk5bS345*) control (*Figure 6B*). To assess the specificity of the reduction effect for this serine, we also systematically substituted adjacent serines with alanines and subsequently measured channel activity (*Figure 6—figure supplement 1A*). While *kcnk5bS345A* showed reduction in activity, the substitution of other serines did not (*Figure 6C*, *Figure 6—figure supplement 1A–D*).

To determine whether the activity of the Kcnk5b channel is associated with the phosphorylation state of this serine, we exchanged the serine for a glutamic acid in order to mimic serine phosphorylation (*kcnk5bS345E*) (*Figure 6C*). Expression of this mutant displayed elevated $K^+$ conductance compared to *knck5bS345* wild-type channel (*Figure 6D*). In addition, the *kcnk5bS345E* mutant was resistant to calcineurin-mediated inhibition (*Figure 6D*). Moreover, the substitution of other serines with glutamic acid had no effect on channel activity, and calcineurin could still regulated the channel (*Figure 6—figure supplement 1A,E,F*). Together, these results indicate that S345 is the important post-translational regulatory serine involved in calcineurin-mediated regulation of Kcnk5b activity.

To determine whether there is a functional relationship between fin scaling and S345-mediated Kcnk5b channel activity, we placed the cDNA of each channel version (wild-type *kcnk5bS345*, *kcnk5bS345E*, or *kcnk5bS345A*) under the control of the heat-shock inducible *hsp70* promoter to generate conditionally inducible transgenes for in vivo expression in the fish. We heat shocked the caudal fins of non-transgenic and transgenic fish lines once daily and subsequently measured the length of each fin in relation to the length of each body (fin-to-body ratio). Non-transgenic siblings were included in the same heat-shock regimen to serve as controls. After 12 days of the heat-shock regimen, we noticed differences between the rates of the regenerating caudal fins lobes of the different transgenic lines (*Figure 6E*) after standardizing the length of each fin to the length of the body (*Figure 6—figure supplement 1G*). By assessing regenerating lobe-to-body measurements over time, we observed that Tg[*hsp70:kcnk5bS345E*] fish maintained the highest rates of allometric regenerative growth, while Tg[*hsp70:kcnk5bS345A*] displayed the lowest growth rates of the transgenic lines (*Figure 6E*). There was no significant change in the rates of growth of the bodies (*Figure 6—figure supplement 1G*). We also assess the final proportional size of unamputated fin lobes between the different transgenic lines, and we observed a linear relationship between the proportional length of the unamputated lobes and the putative phosphorylation status of the channels: the S345A dephosphorylation mimic displayed the smallest growth proportions (*Figure 6I*), the S345E phosphorylation mimic displayed the largest growth proportions (*Figure 6I*), and the average value of the wild-type regulatable version of the channel was between the highly active S345E and marginally active S345A mutants (*Figure 6I*).

To determine whether similar growth effects are generated by other two-pore $K^+$ leak channels, we generated a transgenic fish line that overexpresses *kcnk9*, Tg[*hsp70:kcnk9*-GFP]. Compared to transgenic fish that were not heat-shocked (*Figure 6Ja,K*) and heat-shocked of non-transgenic fish (*Figure 6K*), we observed that heat-shock-induced overexpression of *kcnk9* in the adult fins increased fin outgrowth (*Figure 6Jb,K*; *Figure 6—figure supplement 1H*), which further supports

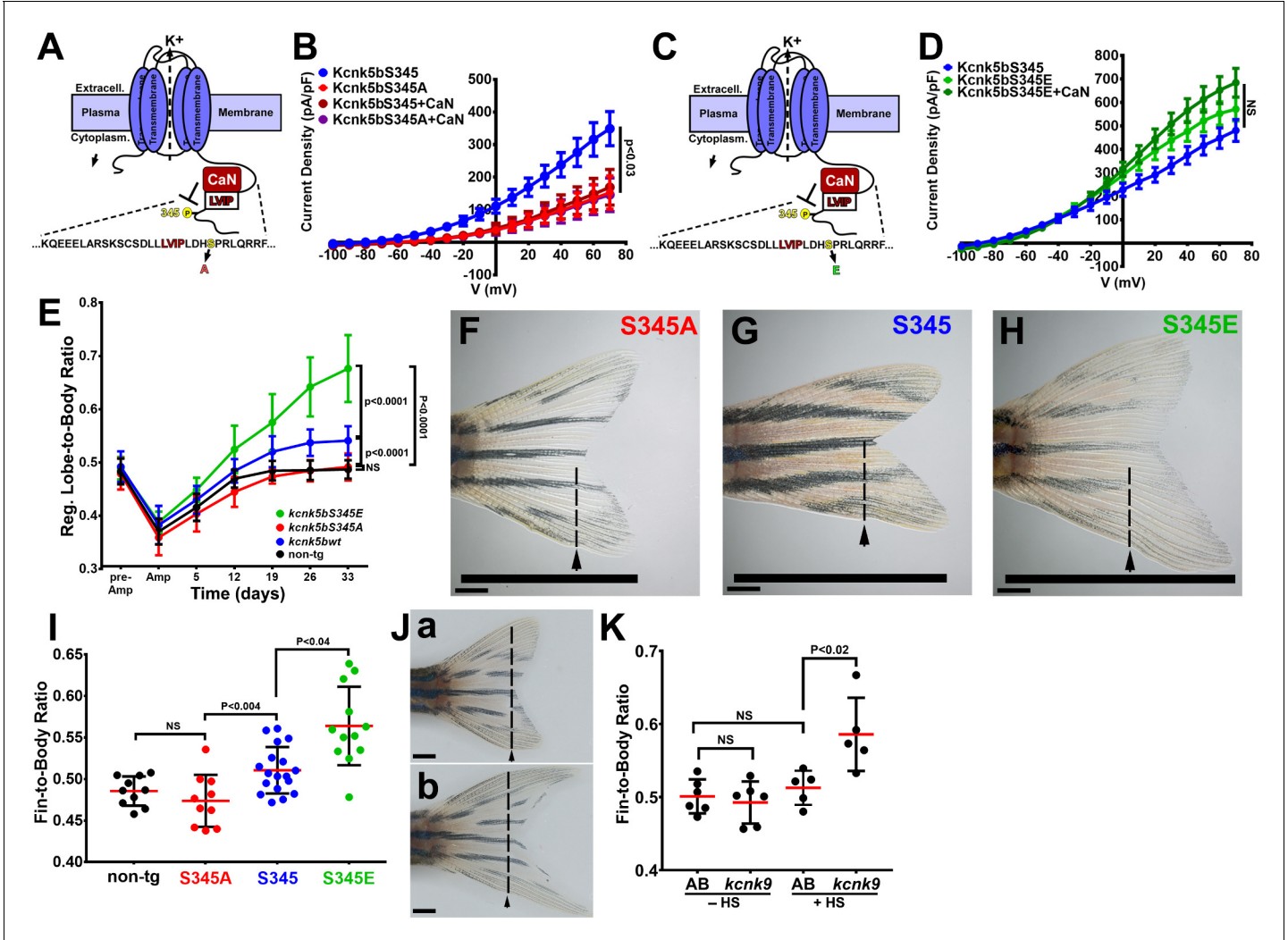

**Figure 6.** Regulation of Kcnk5b controls scaling of the fin. (A) Diagram of Kcnk5b channel showing proposed Serine345Proline346 calcineurin dephosphorylation site adjacent the calcineurin-interaction site (LVIP). Mutation of S345 to alanine (A) mimics dephosphorylation. (B) Whole-cell patch-clamp results of HEK239T (HEK) cells transfected with zebrafish wild-type channel (Kcnk5bS345) or the dephospho-mimic mutant (KcnkS345A) either with or without calcineurin (CaN). (C) Diagram of serine (S) to glutamic acid (E) substitution to mimic phosphorylation of Kcnk5b. (D) Whole-cell patch-clamp measurements for wild-type Kcnk5b and mutant Kcnk5b harboring a Serine345 to glutamic acid either with or without calcineurin (CaN). (E) Graph displays different growth rates of the regenerating caudal fin lobes of the indicated transgenic fish lines. Body length of each fish was used to standardize the fin length measurements (fin-to-body ratio). (F) Caudal fin of Tg[*hsp70:kcnkbS345A*] transgenic fish after regeneration of ventral lobe. (G) Caudal fin of Tg[*hsp70:kcnk5bS345*] transgenic fish after regeneration of ventral lobe. (H) Caudal fin of Tg[*hsp70:kcnk5bS345E*] transgenic fish after regeneration of ventral lobe. (I) Graph of fin-to-body ratios of the unamputated lobes of the indicated transgenic fish lines at 33 days of the same fish as in (F). (J) Representative caudal fins of the Tg[*hsp70:kcnk9*-GFP] fish that was allowed to regenerate without any heat-shock induction of the transgene (a) or underwent a daily 10 min heat-shock induction of the Tg[*hsp70:kcnk9*-GFP] transgene (b). (K) Assessment of regenerative fin growth at 33 dpa of the indicated fish lines and heat-shock treatment. The data for each experiment represent three or more experiments. The averaged data points in B,D,E represent two or more technical replicates per experiment. The graphed data in I,K show all technical replicates. The electrophysiology measurements (panels B,D) are represented as averages with SEM. Significance values shown in the graphs were measured by students t-tests (E, I, K) are represented as averages and SD. The scale bars equal 2 mm (F–H,Ja,b). Arrows and dashed lines in F-H,Ja,b indicate amputation planes through fins.

The online version of this article includes the following source data and figure supplement(s) for figure 6:

**Source data 1.** Electrophysiological data and fin/body measurements.

**Figure supplement 1.** Activity measurements of Kcnk5b Serine mutant channels.

**Figure supplement 1—source data 1.** Electrophysiological data and fin/body measurements.

the conclusion that Kcnk5b-mediated growth is due to its regulation of its electrochemical properties.

Our ability to control the rate of growth by mimicking a specific post-translational modification that can be mediated by calcineurin and that correspondingly determines the level of Kcnk5b activity supports the conclusion that calcineurin regulation of Kcnk5b is an in vivo electrophysiological mechanism through which controlling the potassium conductance of cells scales a vertebrate appendage.

## Discussion

Anatomical structures consist of a combination of different tissue types that develop and grow in a coordinated manner. Recent discoveries show that K$^+$ channels regulate the scaling of fish appendages, but it is still unclear how this electrophysiological signal controls several diverse developmental phenomena within this anatomical structure to achieve coordinated developmental growth. Our results reveal that this in vivo electrical signal has a hierarchical effect on specific developmental genes in the fish fins and larva.

Two-pore K$^+$-leak channels such as Kcnk5b allow K$^+$ to cross the membrane to establish an electrochemical equilibrium, this activity directly affects the membrane potential of the cell (*Goldstein et al., 2001*). Normally, the concentration of K$^+$ is higher on the cytoplasmic side of the plasma membrane due to continual active transport of K$^+$ into the cell by the ATP-dependent Na$^+$/K$^+$ pumps (*Shattock et al., 2015*). As a leak channel, opening of Kcnk5b causes a flow of K$^+$ out of the cell, which hyperpolarizes the membrane potential (*Goldstein et al., 2001*). The previous findings that mutations that increase Kcnk5b channel activity are associated with maintaining allometric growth (*Perathoner et al., 2014*) argue that such changes in membrane potential promote disproportional growth. Our current findings further these previous findings by showing that conditional induction of Kcnk channel activity is sufficient to induce morphogen pathways (*Figures 1*, *2* and *4*) in different in vivo and in vitro contexts, demonstrating transcriptional control of particular developmental mechanisms by different two-pore K$^+$-leak channels.

In addition to K$^+$-leak channels, cells regulate intracellular K$^+$ through different channels and exchangers. Inward rectifying K$^+$ channels allow K$^+$ to enter the cell along the ion's electrochemical gradient. Exchangers will exchange K$^+$ with different substrates (e.g. Na$^+$) to facilitate the entry or removal of K$^+$ based on the concentration gradient of K$^+$ and the exchanged substrate. Previous findings show the importance of the inward rectifying K$^+$ channel Kir2 for cranial-facial and digit defects in humans (*Andersen et al., 1971*; *Canún et al., 1999*; *Sansone et al., 1997*; *Tawil et al., 1994*; *Yoon et al., 2006a*; *Yoon et al., 2006b*). Knockout of the mouse Kir2 channels results in similar head and digit defects (*Zaritsky et al., 2000*), and dominant-negative inhibition of the *Drosophila* Kir2 leads to wing appendage defects that are analogous to the human and mouse appendage defects (*Dahal et al., 2012*). While the mammalian phenotypes remain unexplained, the defects in the *Drosophila* wings have been linked to reduced Dpp (BMP) signaling (*Dahal et al., 2012*), suggesting that intracellular K$^+$ homeostasis is important for BMP signaling.

Removal of an ATP-sensitive K$^+$ channel in the early *Xenopus* embryo disrupts eye formation, while ectopic expression of this channel will produce ectopic eyes in the head and in locations that were not considered to be competent for producing eyes (*Pai et al., 2012*). The ability to ectopically generate eyes was linked to electrophysiological hyperpolarization of the cells and the activation of Pax6-eyeless gene (*Pai et al., 2012*), a master regulator for eye development (*Chow et al., 1999*; *Halder et al., 1995*). In planaria, shortly after wounding, membrane depolarization acts as an early anterior signal that is sufficient (even when induced on the posterior side) to promote the consequent formation of all the anterior structures of the planarian head by inducing notum expression, which inhibits β-catenin-dependent Wnt signal transduction (*Durant et al., 2019*). Furthermore, it was recently revealed that in the *Drosophila* wing discs, depolarization events promote membrane localization of the transmembrane receptor smoothened to promote Shh signaling and that membrane potential values are patterned within the wing disc (*Emmons-Bell and Hariharan, 2021*). These discoveries show that electrophysiological changes are important signals in the formation and growth of anatomical structures.

A recent finding showed that calcineurin inhibition or increased activity of the voltage-gated Kcnh2 channel (*lof* mutant) promotes regenerative outgrowth from the proximal side of a mid-fin excavation, a site that normally does not mount a regenerative outgrowth response (*Cao et al.,*

*2021*). Our findings help explain how calcineurin-regulated and electrophysiological changes from potassium channels can lead to broader tissue organizing phenomena by showing the inductive effect that increasing $K^+$ conductance (and calcineurin's regulation of $K^+$ conductance) can have on a broad number of developmental pathways, which is important for coordinating the organized formation of tissues and organs. Furthermore, we posit that the effect of the activity of the Kcnk5b channel is broader than the traditional mechanisms of growth factor/morphogen signaling pathways, because it is not confined to specific signal transduction mechanisms; rather, it has variable broad effects, such as activation of several developmental signals in the adult fin (*Figure 1*), the larva (*Figure 2*) and different mammalian cell lines (*Figure 4*). While we observed that Kcnk5b is sufficient to promote *lef1*-mediated transcription (*Figures 1–3*), we did not observe direct activation of *axin2*, a down-stream target gene of β-catenin activity (*Figure 1—figure supplement 1C*). We posit that Kcnk5b activity does not directly activate canonical Wnt, but primes cells for increased β-catenin activity upon the reception of a Wnt signal through up-regulation of Lef1. Ultimately, continued Kcnk5b activity must lead to increased β-catenin-dependent Wnt signaling, since all evidence indicates that Wnt signaling is required for allometric fin outgrowth (*Kawakami et al., 2006*; *Stoick-Cooper et al., 2007*), and allometric fin growth is the phenotype that we get by *kcnk5b* or *kcnk9* overexpression in amputated and unamputated fins (*Figure 6E–L*). We propose that the competence to activate different developmental pathways by electrophysiological changes is because the responding cells are either primed to activate them or the pathways are already active. It will be important to find out how this electrophysiological signal coordinates the activity of these developmental signals. In this regard, only few factors are known that regulate *shh* and *lef1* transcription. Thus, our finding that an electrophysiological mechanism is involved not only provides a new understanding of how electrophysiology acts as an inductive signal, it also may lead to the discovery of molecular mechanisms that control the expression of these mediators of important morphogen signals.

The scaling activity of Kcnk5b includes all the tissues of the entire appendages of the fish (*Perathoner et al., 2014*). Previous findings implicate broader intercellular electrophysiological gradients as a mechanism for tissue growth (*Adams and Levin, 2013*). Electrophysiological measurements of animal tissues show that electric fields are generated and are important in vivo (*Borges et al., 1979*; *Jenkins et al., 1996*; *McGinnis and Vanable, 1986*), which suggests the existence of in vivo bioelectric information that regulates physiological phenomena. However, from our transplantation experiments, we observe that Kcnk5b's effect is cell autonomous (*Figure 3*). Consequently, the question arises about how the activity of this $K^+$-leak channel relates to a broad, coordinated phenotype of scaling the several tissues of the fin. An answer is that the autonomous transcriptional programs include morphogens. What is unclear is whether a limited number of cells in the fin control the growth and organizing information so that Kcnk5b only needs to act on a limited number of cell types, or whether Kcnk5b regulates proportional growth at multiple levels and that the cell autonomous transcriptional response that we observe is one outcome of a combination of intracellular and intercellular responses induced by Kcnk5b.

Changes in membrane potential from alterations in $K^+$ conductance are also associated with the progression through the cell cycle (*Blackiston et al., 2009*; *Urrego et al., 2014*), because $K^+$ channel activity increases at specific cell cycle phases (*Urrego et al., 2014*), and inhibition of $K^+$ channel activity leads to cell cycle arrest in many different tissue cell types (*Blackiston et al., 2009*). It is possible that this phenomenon explains part of Kcnk5b's ability to promote allometric growth. We do not yet know whether other phenomena linked to the activity of mammalian Kcnk5 [influence cell tonicity (*Niemeyer et al., 2001*), metabolic acidosis and alkalinization (*Warth et al., 2004*), $CO_2/O_2$ chemosensing in retrotrapezoid nucleus neurons (*Flores et al., 2011*) and apoptosis in lymphocytes and neurons (*Göb et al., 2015*; *Nam et al., 2011*)] are involved in appendage scaling.

We previously showed that calcineurin inhibition shifts isometric growth to allometric growth (*Kujawski et al., 2014*). Subsequently, Daane et al. showed that this effect is reversible in that removal of calcineurin inhibitors restores isometric growth. The authors also suggest an alternative mechanism for calcineurin regulation of Kcnk5b in its C-terminus (*Daane et al., 2018*); however, their model posits that active calcineurin promotes the activity of Kcnk5b, which does not yet explain how calcineurin inhibition and increased Kcnk5b activity each promote allometric growth (*Kujawski et al., 2014*; *Perathoner et al., 2014*). In either case, these data implicate calcineurin as a molecular switch governing isometric versus allometric growth control. Our findings provide a

mechanism for how this switch acts to scale the fish appendages by directly regulating the activity of Kcnk5b through the dephosphorylation and phosphorylation of a specific serine (*Figure 7*). The ability to mimic or block calcineurin regulation of this $K^+$-leak channel (*Figure 5*), whose activity levels directly translate into the extent of allometric growth (*Figure 6*), defines how calcineurin inhibition expands clonal populations during fin regeneration (*Tornini et al., 2016*). However, as we observed from both calcineurin inhibition (*Kujawski et al., 2014*) and from conditionally inducing Kcnk5b activity (*Figure 6*), the induced allometric growth of the entire fin is more than expanding clonal populations, since the outcome is not tumorigenesis. Instead, the growth is coordinated among all the tissues (*Figure 6G–I,K,L*; *Kujawski et al., 2014*; *Perathoner et al., 2014*), and our finding that Kcnk5b activates several developmental pathways (*Figures 1*, *2* and *4*) argues that calcineurin activity acting on Kcnk5b regulates more than cell cycle progression.

An important next step is to learn how the calcineurin-Kcnk5b circuit is integrated into the broader mechanisms that scale the appendages. Calcineurin is a $Ca^{2+}$-dependent enzyme which suggests that intracellular $Ca^{2+}$ is involved in scaling information. $Ca^{2+}$ is a broad second messenger that can activate several downstream $Ca^{2+}$-dependent enzymes, so broad changes in its subcellular levels likely have multiple effects. It remains unclear whether there is a specific intracellular

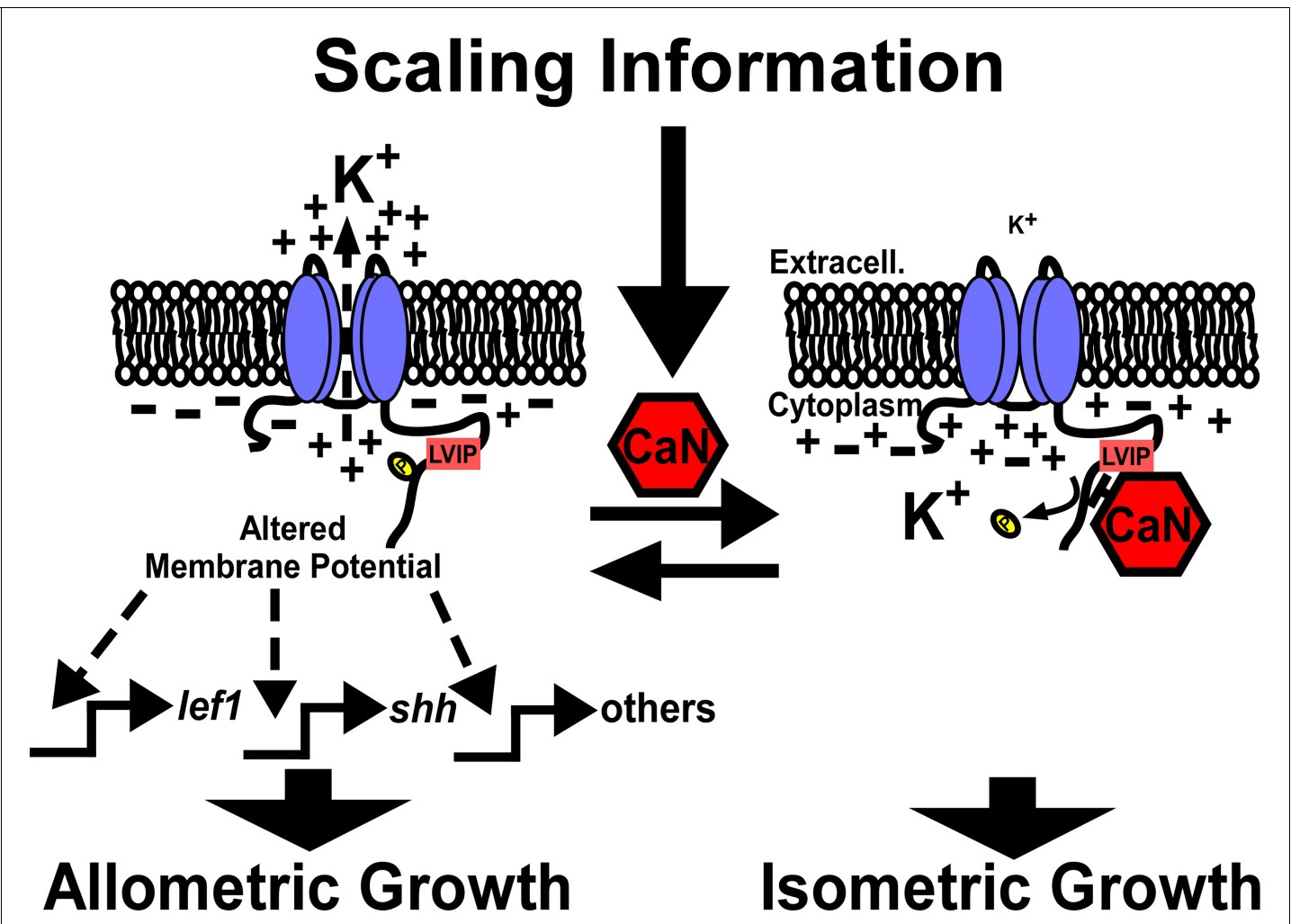

**Figure 7.** Model of calcineurin regulation of Kcnk5b-mediated activation of developmental gene transcription. Kcnk5b activation results in reduced cytoplasmic $K^+$, which is sufficient to induce the transcription of the *shh* ligand and *lef1* transcription factor as well as components of other developmental pathways to induce coordinated allometric growth of the tissues of the fish fin appendage. Scaling information from the body or local tissues in the fin activate calcineurin so that it dephosphorylates Kcnk5b on S345 to reduce its $K^+$-channel activity, which results in isometric growth of the fin.

distribution pattern that leads to calcineurin-mediated control of scaling. It is also possible that Ca$^{2+}$-mediated activation of calcineurin—and consequent restoration of isometric growth—is so dominant that other Ca$^{2+}$-mediated activities have little effect.

Two mechanisms that regulate proportional growth of organs are vitamin D and Hippo signaling. Increasing vitamin D signaling enhance the growth of the entire body, including the fins (*Han et al., 2019*). We propose that vitamin D is a systemic body signal that ultimately leads to the increase in Kcnk5b signaling. It is also possible that this hormone acts independently of Kcnk5b. In *Drosophila*, the Hippo pathway regulates brain size and size of the imaginal discs (*Poon et al., 2016*; *Rogulja et al., 2008*). Mice overexpressing a nuclear version of the Hippo-signaling component Yap1 in the adult liver develop significantly enlarged livers (*Camargo et al., 2007*; *Dong et al., 2007*). The Hippo signal transduction pathway consists of several core components that can be regulated by different factors at plasma membrane and within the cell (*Yu and Guan, 2013*), so there are several possible nodes of interaction between of Kcnk5b and Hippo cascade. It is also possible the Hippo-mediated transcription regulates *kcnk5b* expression or channel activity.

Connexin43 also regulates proportional growth of the fins, since mutations that reduce the intercellular connectivity of connexin43 produce adult fins that are half the size as the fins of wild-type siblings (*Hoptak-Solga et al., 2007*; *Iovine et al., 2005*). The connective nature of these intercellular junction proteins indicate that direct communication between intracellular compartments of tissue cells is an important component of the scaling mechanism of the fins. Our observation that Kcnk5b cell-autonomously activates the *7XTCF-Xla.sam*:mCherry reporter (*Figure 3*) indicates that it is not due to intracellular transfer of K$^+$. It is still unclear whether the disruption of intracellular trafficking of other ions (such as Na$^+$ or Ca$^{2+}$) or of other factors is responsible for the connexin43's effect on scaling.

Kcnk5b's ability to activate the Lef1-dependent reporter cell autonomously in different tissue types supports the conclusion that K$^+$ conductance has the potential to regulate developmental transcription in a broad range of tissues (*Figure 3B–E*). The observation that neuronal cells in the brain and myocytes in the trunk muscle respond similarly to non-excitable cells elsewhere in the body suggests that even cells that harbor action potentials use K$^+$ conductance to regulate gene expression. However, when we either stably expressed or transient transfected Kcnk5b in different cell lines, we did not observe consistent activation of Lef1 or Shh (*Figure 4*). We posit that the competencies of the cultured cell lines are different from the developmental competencies of the in vivo cells. Also, the mammalian cell lines are cancer cells, so they may already have an increased transcriptional base line for the selected developmental genes that the two-pore potassium-leak channels can only partially affect.

While we observed similarities between Kcnk5b and Kcnk9 or Kcnk10 in HEK cells, in Hela and N2A cells, we observed only partially similar profiles for the selected genes (while Kcnk9 and Kcnk10 were similar) (*Figure 4*). We postulate that the observed differences may come from one or both of the following explanations. First, different responses from channels of the same ion-type are due to different levels of membrane potential changes, which results in different levels of gene transcription. Second, these channels have different intracellular sequences, which may determine other unknown intermolecular interactions that have different signal transduction properties. In any case, we did observe that transgenic overexpression of Kcnk9 produces a similar allometric growth of the caudal fin (*Figure 6K,L*). It remains to be explored whether intermolecular interactions between the channels and another protein contribute to the scaling of other organs as well as how other electrophysiological mechanisms that control membrane potential have the same growth effect.

In conclusion, we show how a specific electrophysiological mechanism activates important morphogen pathways to scale tissues in different in vivo contexts. We propose the observed diversity in morphogen and growth factor expression to Kcnk5b activity explains why the increased activity of Kcnk5b produces the diverse transcriptional response in the different tissues associated with the observed coordinated outgrowth of the entire fin. Also, we show how changes in phosphorylation of S345 in the cytoplasmic C-terminus can be regulated by calcineurin to directly control electrophysiological activity of the channel to scale the fin. Thus, we offer an in vivo paradigm in which membrane potential acts as potent regulator of coordinated developmental signaling, and we show how the two-pore K$^+$-leak channel Kcnk5b is involved in the initial activation of specific developmental mechanisms that lead to the scaling of the fish fin appendages.

# Materials and methods

## Key resources table

| Reagent type (species) or resource | Designation | Source or reference | Identifiers | Additional information |
|---|---|---|---|---|
| Gene (*Danio rerio*) | Kcnk5b | genebank | ZFIN:ZDB-GENE-040426–1297 | |
| Gene (*Danio rerio*) | Kcnk9 | genebank | ZFIN:ZDB-GENE-070705–260 | |
| Gene (*Danio rerio*) | Kcnk10a | genebank | ZFIN:ZDB-GENE-041210–291 | |
| Gene (*Homo sapiens*) | CaN | genebank | HGNC:HGNC:9314 | |
| Gene (*Danio rerio*) | lef1 | genebank | ZFIN:ZDB-GENE-990714–26 | |
| Gene (*Danio rerio*) | shha | genebank | ZFIN:ZDB-GENE-980526–166 | |
| Gene (*Danio rerio*) | aldh1a2 | genebank | ZFIN:ZDB-GENE-011010–3 | |
| Gene (*Danio rerio*) | pea3 | genebank | ZFIN:ZDB-GENE-990415–71 | |
| Gene (*Danio rerio*) | msxb | genebank | ZFIN:ZDB-GENE-980526–312 | |
| Gene (*Danio rerio*) | ptch1 | genebank | ZFIN:ZDB-GENE-980526–196 | |
| Gene (*Danio rerio*) | ptch2 | genebank | ZFIN:ZDB-GENE-980526–44 | |
| Gene (*Danio rerio*) | bmp2b | genebank | ZFIN:ZDB-GENE-980526–474 | |
| Gene (*Danio rerio*) | axin2 | genebank | ZFIN:ZDB-GENE-000403–2 | |
| Gene (*Danio rerio*) | bactin2 | genebank | ZFIN:ZDB-GENE-000329–3 | |
| Gene (*Homo sapiens*) | lef1 | genebank | HGNC:HGNC:6551 | |
| Gene (*Homo sapiens*) | shh | genebank | HGNC:HGNC:10848 | |
| Gene (*Homo sapiens*) | aldh1a12 | genebank | HGNC:HGNC:15472 | |
| Gene (*Homo sapiens*) | pea3 | genebank | HGNC:HGNC:3493 | |
| Gene (*Homo sapiens*) | msxb | genebank | HGNC:HGNC:7391 | |
| Gene (*Homo sapiens*) | GAPDH | genebank | HGNC:HGNC:4141 | |
| Strain, strain background (AB as background, both male and female) | Tg[hsp70:Kcnk5b-GFP] | AB fish from CZRC | AB fish Catlog ID: CZ1 | Based on tol2 transposable element |
| Strain, strain background (AB as background,both male and female) | Tg[hsp70:Kcnk5bS345A-GFP] | AB fish from CZRC | AB fish Catlog ID: CZ1 | Based on tol2 transposable element |
| Strain, strain background (AB as background,both male and female) | Tg[hsp70:Kcnk5bS345E-GFP] | AB fish from CZRC | AB fish Catlog ID: CZ1 | Based on tol2 transposable element |
| Strain, strain background (AB as background,both male and female) | Tg[hsp70:Kcnk9-GFP] | AB fish from CZRC | AB fish Catlog ID: CZ1 | Based on tol2 transposable element |
| Strain, strain background (AB as background,both male and female) | Tg[7XTCF-Xla.sam:mCherry] | European Zebrafish Resource Center | Cat.#:15200 | |
| Genetic reagent (Oryzias latipes) | Transposase RNA | This paper | This paper | In vitro transcripted by sp6 kit |
| Genetic reagent | DIG RNA Labeling Mix | Roche | Cat.#:1277073 | |
| Genetic reagent (*E. coli*) | T7 RNA-polymerase | Promega | Cat. #:P207B | |
| Cell line (human) | pLenti-CMV-Kcnk5b-EGFP | OBiO | Contract number: HYKY-181108018-DLV | Titer:1.67E*08 TU/ml |

*Continued on next page*

*Continued*

| Reagent type (species) or resource | Designation | Source or reference | Identifiers | Additional information |
|---|---|---|---|---|
| Cell line (human) | pLenti-CMV- EGFP | OBiO | Contract number: HYKY-181108018-DLV | Titer:1.55E*09 TU/ml |
| Transfected construct (human) | CMV-Kcnk5b-GFP | This paper | Primers from https://www.genewiz.com.cn | Kcnk5b is cloned from adult zebrafish fin cDNA library |
| Transfected construct (human) | CMV-Kcnk9-GFP | This paper | Primers from https://www.genewiz.com.cn | Kcnk9 is cloned from zebrafish 3dpf larva cDNA library |
| Transfected construct (human) | CMV-Kcnk10-GFP | This paper | Primers from https://www.genewiz.com.cn | Kcnk10 is cloned from zebrafish 3dpf larva cDNA library |
| Transfected construct (human) | CMV-CaN-Mcherry | This paper | Primers from https://www.genewiz.com.cn | CaN is cloned from adult zebrafish fin cDNA library |
| Transfected construct (human) | CMV-Kirin | *Shen et al., 2019* | | |
| Biological sample (zebrafish) | Adult zebrafish fin tissue | This paper | This paper | Freshly isolated after heatshock at indicated time points |
| Antibody | Anti-GFP (mouse monoclonal) | Invitrogen | Cat. #: MA5-15349 RRID:AB_987186 | IF(1: 400) |
| Antibody | Anti-β-catenin (rabbit polyclonal) | Cell Signaling | Cat. #: 9562L RRID: B_331149 | WB(1:1000) IF(1:400) |
| Antibody | Anti-Lef1(rabbit monoclonal) | Cell Signaling | Cat. #:2230T RRID:AB_823558 | WB (1:1000) |
| Antibody | Anti-Shh(rabbit polyclonal) | Novus | Cat. #: NBP2-22139 | WB (1:1000) |
| Antibody | Anti-mCherry antibody(rabbit polyclonal) | Invitrogen | Cat.#:PA534974 RRID: AB_2552323 | IF(1:1000) |
| Antibody | Anti-mouse-GFP (goat polyclonal) | Abcam | Cat.#:ab150113 RRID:AB_2576208 | IF(1:1000) |
| Antibody | Goat-anti-rabbit-mCherry secondary antibody (Goat polyclonal) | Abcam | Cat.#:ab150078 RRID:AB_2722519 | IF(1:2000) |
| Antibody | Anti-Mouse IgG (goat Polyclonal) | Sigma | Cat. #:A3682-1ML RRID:AB_258100 | WB(1:80000) |
| Antibody | Anti-Rabbit IgG (H+L) (donkey Polyclonal) | Jackson | Cat. #:711-036-152 RRID:AB_2340590 | WB(1:20000) |
| Antibody | Anti-β-Actin pAb-HRP-DirecT (rabbit Polyclonal) | MBL | Cat. #:PM053-7 RRID:AB_10697035 | WB(1:2000) |
| Antibody | Anti-Digoxigenin-AP, Fab fragments (sheep polyclonal) | Roche | Cat.#:11093274910 | Insitu(1:5000) |
| Recombinant DNA reagent | NovoRec plus One step PCR Cloning Kit | novoprotein | Cat. #:NR005-01B | |
| Sequence-based reagent | F-kcnk9 | This paper | Pcr primers | ATGAAGAGGCAGAACGTGCGGACGC |

*Continued on next page*

*Continued*

| Reagent type (species) or resource | Designation | Source or reference | Identifiers | Additional information |
|---|---|---|---|---|
| Sequence-based reagent | R-kcnk9 | This paper | Pcr primers | GATGGACTTG CGTCGTCTCA TAAGCCGG |
| Sequence-based reagent | F-kcnk10 | This paper | Pcr primers | ATGAAATTTCCA ACGGAAAACCC GAGGAAG |
| Sequence-based reagent | R-kcnk10 | This paper | Pcr primers | CTATGGATCCAC CTGCAAACGGAACTC |
| Commercial assay or kit | Protein Quantitative Kit (BCA) | MDBio | Cat. #:KT054-200rxn | |
| Chemical compound, drug | FK506 | Sigma | Cat. #:F4679-5MG | |
| Chemical compound, drug | DL-Dithiothreitol | Promega | Cat. #:P117B | |
| Chemical compound, drug | Adenosine 5'-triphosphate disodium salt hydrate | sigma | Cat. #:A2383 | |
| Chemical compound, drug | Nitro Blue Tetrazolium | Sigma | Cat. #:N6639 | |
| Chemical compound, drug | BCIP | Sigma | Cat. #:B-8503 | |
| Chemical compound, drug | DAPI | Roche | Cat.#:10236276001 | |
| Software, algorithm | ImageJ | Open source | RRID:SCR_003070 | https://imagej.nih.gov/ij/ |
| Software, algorithm | SymPhoTime 64 | PicoQuant | RRID:SCR_016263 | Used to analysis flim data |
| Software, algorithm | Zen Blue | Zeiss | RRID:SCR_013672 | |
| Software, algorithm | Patchmaster | Heka | RRID:SCR_000034 | |
| Software, algorithm | Graphpad prism | Graphpad Software | RRID:SCR_002798 | |
| Software, algorithm | Clampfit10.5 | Axon | RRID:SCR_011323 | |
| Other | ExpressPlus PAGE Gel, 8–16%, 15 wells | Genscript | Cat.#:M81615C | For western blotting |
| Other | Western Lightning Plus ECL 680 | PerkinElmer(PE) | Cat.#:NE L105001EA | For western blotting |
| Other | MMESSAGE MMACHINE SP6 KIT | Invitrogen | Cat.#:AM1340 | |

## Cloning

Constructs were designed either using standard restriction enzyme or by homologous recombination methods. *kcnk5b* cDNA was isolated by RT-PCR from regenerating adult fin cDNA library and

cloned into MCS region of pcDNA6-myc-6xHIS-tag plasmid (Invitrogen) or pBluescript harboring the *hsp70* zebrafish promoter and GFP coding sequence surrounded by two miniTol2 sites. Mutagenesis of the Serine345 codon of Kcnk5b was performed using QuikChange Mutagenesis kit (Agilent).

## Zebrafish husbandry

AB strain fish were raised in 10L tanks with constantly flowing water, 26℃ standard light-dark cycle (*Brand et al., 2002*) in either a Schwarz (DFG-Center for Regeneration, TU Dresden) or HaiSheng aquarium (ShanghaiTech University) systems. Transgenic lines harboring the different *kcnk5b* transgenes were created by injecting 300 µg of each construct together with mRNA of Tol2 transposase (*Balciunas et al., 2006*). Fish harboring Tg[7*XTCF-Xla.sam*:mCherry] transgene are deposited in the European Zebrafish Resource Center. Fish embryos and larva were raised in 1xE3 medium (5 mM NaCl, 0.17 mM KCl, 0.33 mM CaCl$_2$, 0.33 mM MgSO$_4$, 10$^{-5}$% Methylene Blue) until 10-12dpf, then transferred to aquarium water tanks to grow. Transgenic lines established by screening for GFP expression after heat shock. Experiments used male and female fish equally. Fish experiments were compliant to the general animal welfare guidelines and protocols approved by legally authorized animal welfare committees (Technische Universität Dresden, Landesdirektion Dresden, and ShanghaiTech University, ShanghaiTech Animal Welfare Committee).

## Heat-shock induction of transgenes

Parents of heat-shock-driven transgenic lines were either outcrossed to same-strain wild-type fish or to fish harboring Tg[7*XTCF-Xla.sam*:mCherry]transgene. Progeny were collected in 1xE3 and raised at 28℃. Carriers were confirmed positive for their respective transgenes by a single heat shock at 37℃ for 1 hr. For embryo experiments, heat shock was at 12 hpf or 32 hpf (pectoral fin bud) in 37℃ E3 medium for 30 min. For larva the heat shock was in 37℃ E3 medium for 30 min at 2 dpf. For adult fin, 6-month-old fish underwent a daily heat-chock regimen: first, sedated in 0.04% tricane in aquarium water, then placed in conical tubes containing 0.04% tricane in aquarium water to allow continued gill movement in oxygenated water and allow the caudal fins to be exposed to 37℃ water for 7 min. After heat shocking the caudal fin, the fish were returned to flowing aquarium water and monitored daily. Caudal fin measurements were made from the base of the fin to the distal tip along the third fin ray of each fin lobe counting in from the dorsal-most and ventral-most sides. Body lengths were determined by measuring the most anterior point of the jaw to the base of the caudal fin. Larval finfold measurements were from the ventral-most to dorsal-most points of the caudal finfold or from the anterior-most to the posterior-most points of the ventral finfold along the body length. Body lengths of larva were measured from the anterior-most point of the jaw to the posterior-most point of the somatic musculature.

## Western blotting

Fins were amputated and snap frozen in liquid nitrogen and homogenized in RIPA buffer (Proteintech, B100020) with protease inhibitor (Pierce, 88666) and then rotated at 4℃ for 2 hr to lysate cells thoroughly. Protein was collected and concentration was quantified by BCA assay (MDBio, KT054-200rxn) after centrifuge at 12000 rpm at 4℃ for 20 min. Then gel electrophoresis was conducted at 120V for 90 min using page gels (Genscript, M81615C), protein were then transferred to PVDF membranes (Bio-Rad, 1620177). Anti-lef1 (1:1000, Cell Signaling Technology, 2230T, (RRID: AB_823558)), Anti-shh (1:1000, Novus, NBP2-22139, RRID: AB_331149), Anti-β-catenin (1:1000, Cell Signaling Technology, 9562L), Anti-β-Actin pAb-HRP-DirecT (1:2000, MBL, PM053-7, RRID: AB_10697035) were used as primary antibodies to incubate overnight at 4℃. Anti-Rabbit IgG (1:20000, Jackson, 711-036-152, RRID: AB_2340590) were used as secondary antibody and blots were detected using ECL detection (PerkinElmer (PE), NEL105001EA), and Image J software (RRID:SCR_003070) was used for density quantification.

## In situ hydridization

mRNA probes were made from RT-PCR products isolated from two dpf zebrafish embryos 6 hr after 37.5℃ heat shock. The primer sequences for generating the probes are F- *shha*:5'- TGCGGC TTTTGACGAGAGTGC-3'R-shha: 5'-GGTAATACGACTCACTATAGGG TTTCCCGCGCTGTCTGCCG-3' F-lef1: 5'-GAGTTGGACAGATGACCCCTCCTC-3'; R-lef1: 5'-GGTAATACGACTCA CTA

TAGGGGCAGACCATCCTGGGTAAAG-3'. in vitro transcription reagents are from Promega. Isolated fish fins were surgically isolated and incubated in 4% PFA in 1xPBS at 4°C overnight with gentle rocking. Samples were subsequently washed 5 times in 1xPBS and then dehydrated by incubation for 15 min in a graded series of increasing methanol/1xPBS solutions (25%, 50%, 75%, 3 × 100%) on ice. Fins were then incubated in 100% methanol for ≥2 hr at −20°C. Samples were then rehydrated using the reversed dehydration series of (methanol/1xPBS solutions). Samples were then incubated more than 4x in 1xPBS to remove all methanol, and subsequently incubated in 10 µM Proteinase K for 10 min at RT. Samples were then incubated 20 min. in 4% PFA/1xPBS to inactivate the Proteinase K. Samples were incubated in 1xPBS 6 × 10 mins to remove the PFA, then incubated in pre-hybridization buffer for 2 hr at 65°C. Samples were subsequently incubated in the hybridization solution containing 200 ng/ml of each mRNA probe ≥14 hr at 65°C. Samples then were washed with successive wash steps to remove unbound probe and prepare for antibody incubation: twice 2xSSC/50% deionized formalin at 65°C, twice 2xSSC/25% deionized formalin at 65°C, 2xSSC at 65°C, twice 0.2xSSC at RT, 6 times 1xPBST (1xPBS with Tween-20), once in blocking solution (2% Bovine albumin [Sigma-Aldrich, A3294-100G], 2% Sheep Serum [Meilunbio, M134510]) at RT for 4 hr. Samples were incubated with Anti-digoxigenin-AP Fab Fragment (Sigma-Aldrich, 11093274910, RRID: AB_514497) in blocking solution ≥14 hr at 4°C. Samples were then washed 6 times with 1xPBST, subsequently incubated in [0.1 M Tris-HCl, pH 9.5, 0.1 M NaCl, 0.05 M MgCl$_2$] 3 times for 30 min, and then in Nitro Blue Tetrazolium (Sigma-Aldrich, N6639-1G) and 5-bromo-4-chloro-3-indolyl phosphate (Sigma-Aldrich, 136149) in [0.1 M Tris-HCl, pH 9.5, 0.1 M NaCl, 0.05 M MgCl$_2$] at RT ≥8 hr. Samples images under Stemi508 stereoscope (Zeiss) with Axiocam ERc5s digital color camera (Zeiss) and Zen2.3 software (Zeiss, RRID: SCR_013672). For signal areas and signal intensities analyses, images of the *shh* in situs were transferred to Fiji ImageJ (RRID:SCR_003070). The stained regions were traced using 'free ROI' and then quantitated using the 'measure' function under the 'analysis' menu to calculate the number of pixels contained in the stained area. For signal intensity analysis, the signal region and adjacent unstained regions were measured by 'free ROI' selection and the mean intensity pixel values were determined from the 'measure' function under the analysis menu by subtracting the mean value of the unstained region from the mean value of the signal region for each fin bud.

## Immunohistochemistry

Zebrafish fins or larvae euthanized in 1% Mesab were fixed in 4%PFA/1xPBS and embedded in 1% agarose (CryoStar NX50, Thermofisher) or tissue freezing medium (Leica, 14020108926) on dry ice. (CryoStar NX50, Thermofisher) before cryosectioning. 10 µm sections were mounted on glass slides (Titan, 02036398) and dried. The tissue freezing medium (Leica) was removed in ddH$_2$O for 10 min. Sections were permeabilized in 0.1%Triton-X for 5 min and incubated in 1% BSA/1xPBS/0.1%Tween-20 (PBST) at room temperature for 30 min. Sections were incubated in a mouse-anti-GFP (Invitrogen, MA5-15349, RRID:AB_10987186) solution (1:400), Anti-β-catenin (Cell Signaling, 9562L, RRID: AB_331149), Anti-Lef1 (Cell Signaling, 2230T, RRID: AB_823558) solution (1:1000), Anti-Shh (Novus, NBP2-22139, RRID: AB_2883969) solution (1:1000 dilution), rabbit-anti-mCherry antibody (Invitrogen, P5-34974, RRID: AB_2552323) solution (1:2000 dilution) in PBST at 4°C overnight (>12 hr). The primary antibody solution was replaced by a goat-anti-mouse-GFP (Abcam, ab150113, RRID: AB_2576208), goat-anti-rabbit-mCherry secondary antibody (Abcam, ab150078, RRID: AB_2722519) solution (1:2000 dilution) in PBST and incubated at RT in the dark for 60 min. The secondary antibody solution was then replaced with DAPI solution (Roche, 10236276) in the dark at room temperature for 5 min. DAPI were washed away in 1xPBST 3 × 5 min incubations at RT. Coverslips (Titan, 02036401) were mounted with a 40% glycerol solution and sealed with nail polish. The sections were visualized using an LSM710 upright scanning confocal (Zeiss) or a LSM880 inverted scanning confocal (Zeiss) with ZENBlue software (Zeiss, RRID: SCR_013672). Images were processed with Fiji software (RID:SCR_003070).

## β-catenin nuclear analysis on IHC cross-sections of fins

Using Fiji ImageJ, multiple nuclei of cells in IHC fin cross-section image of control fins were manually measured, and a mean mRFP fluorescence intensity value was calculated. This mean value was used as the baseline for assessing nuclear β-catenin levels. β-catenin nuclear values for all the nuclei in

each cross-section were assessed with ImageJ by splitting the combined images of β-catenin and DAPI and using DAPI to have Fiji ImageJ define and select all nuclei in the image. The nuclear β-catenin levels were determined in selected nuclei by intensity analysis in Fiji ImageJ (RID:SCR_003070), which provided a numeric value for the β-catenin channel in all the nuclei of each section.

## Cell transplantations

Transgenic fish lines [*hsp70:kcnk5b*-GFP] and [7XTCF-Xla.sam:mCherry] were inbred for 20–30 min before embryos were collected in E3. The genotypes of the parents and of the imaged progeny were confirmed to be homozygous for the 7XTCF-Xla.sam:mCherry transgene by qPCR (*Table 1*). Primers for genomic β-actin locus were used as DNA content standardization. The cycle procedure was at 50.0°C for 2 min, 95.0°C for 10 min in the hold step; 95.0°C for 15 s, 55.0°C for 20 s for 40 routine in the elongation step; 95.0°C for 15 s, 60.0°C for 1 min, 95.0°C for 15 s in the melting curve step. The embryos were left to develop in E3 for 3–3.5 hr at $28^0$C until the blastula stage. Embryos were then placed in agarose ridges for easy access under a Zeiss stereomicroscope and cells from the [*hsp70:kcnk5b*-GFP] embryos were isolated by air suction via a glass needle mounted on a Precision Instruments piston. The [*hsp70:kcnk5b*-GFP] cells were then transplanted into the [*7XTCF-Xla.sam*:mCherry] embryos. The transplants were carefully moved from the agarose into clean 90 mm dishes with fresh E3 (±20–25 per dish) and incubated at 28°C. One or two days after transplantation, fish were heat shocked for 1 hr at 37°C. Four hours after heat shock, 48- or 72 hr fish were embedded into 1% low melting agarose (Sigma-Aldrich-Aldrich, A9414-250G) supplemented with Mesab in 35 mm glass bottom confocal dishes (Cellvis: D35-20-1-N) and turned to their side. Visualized with a LSM710 confocal argon laser microscope (Zeiss) with ZENBlue software (Zeiss, RRID: SCR_013672).

## qRT-PCR

The cDNA that was used for qRT-PCR was extracted from the HEK293T cells and zebrafish. The mRNA was isolated using Tri-Reagent (CWBio, 03917). Then 1 $\mu g$ mRNA was used for the reverse transcription to cDNA using 4x gDNA wiper Mix, 5x HiScript III qRT SuperMix (Vazyme, L/N 7E350C9). qRT-PCR was performed using 2x ChamQ Universal SYBR qPCR Master Mix (Vazyme, L/N TE342F9) with QuantStudio3 machine (Thermofisher) with the following primers (*Tables 2*, *3*, *4*)). The cycle procedure was at 50.0°C for 2 min, 95.0°C for 10 min in the stage 1; 95.0°C for 15 s, 60.0°C for 20s for 40 routine in the stage 2; 95.0°C for 15s, 60.0°C for 1 min, 95.0°C for 15s in the Melt Curve. Samples were standardized using the detected β-actin expression for fish mRNA isolation and GAPDH for human cell line mRNA isolations. The data was analyzed in the ΔΔCt method.

## Cell culture

All cell culture lines were incubated at 37°C, 5% $CO_2$, 95% humidity in incubators (Thermofisher, FORMA STERI-CYCLE i160) in DMEM medium (Gibco,1199506) with 10% FBS (Gibco,1009914) and 1% penicillin streptomycin (Gibco, 15140122). The identity of the cell lines was not authenticated, and mycoplasma was not determined. Cell were split to 50% density and transfected with Lipofectamine (Invitrogen, 11668–019) 12 hr later. Expression for the transfected constructs was evaluated by expression of fluorescent marker.

## Mycoplasma test

Two μl of DMEM medium from different cultured cells are freshly taken as template. Primers to detect mycoplasma are used as F: 5'-GGGAGCAAACAGGATTAGATACCCT-3'; R: 5'- TGCACCATCTGTCACTCTGTTAACCTC-3', then standard PCR was conducted (TOYOBO, KOD-401). The

**Table 1.** Primer sequences for qPCR genotyping.

| **F-mCherry** | **GGCCATCATCAAGGAGTTCATGC** |
| --- | --- |
| R-mCherry | GAGGGGAAGTTGGTGCCGC |
| F-β-actin | GAGCTGCAGTCTAAGCTTTGACC |
| R-β-actin | CATTGCCGTCACCTTCACCGTTC |

**Table 2.** qRT-PCR primer sequences for zebrafish genes.

| F-lef1 | AATGATCCCGTTCAAAGACG |
|---|---|
| R-lef1 | CGCTAAGTCTCCCTCCTCCT |
| F-shha | CCACTACGAGGGAAGAGCTG |
| R-shha | GAGCAATGAATGTGGGCTTT |
| F-aldh1a2 | AACCACTGAACACGGACCTC |
| R-aldh1a2 | CTCCAGTTTGGCTCCTTCAG |
| F-pea3 | AGAAGAACCGTCCAGCCATGA |
| R-pea3 | AACATAACGCTCACCAGCCAC |
| F-msxb | ACACTTTGTCGAGCGTTTCGG |
| R-msxb | TCTTGTGCTTGCGTAAGGTGC |
| F-ptch1 | GGGCAGCTAATCTGGAGACGG |
| R-ptch1 | GCGCCTCTACGGTCAAAATG |
| F-ptch2 | TGCCACGCCGCTTTTGCTTT |
| R-ptch2 | GTTTCAATGGCAGCGACCCG |
| F-bmp2a | GCAGAGCCAACACTATCAGGAG |
| R-bmp2a | CCACTTTAATACAGCAGGAGTTACG |
| F-axin2 | GGACACTTCAAGGAACAACTAC |
| R-axin2 | CCTCATACATTGGCAGAACTG |
| F-cmyc | TAACAGCTCCAGCAGCAGTG |
| R-cmyc | GCTTCAAAACTAGGGGACTG |
| F-cyclinD1 | GCCAAACTGCCTATACATCAG |
| R-cyclinD1 | TGTCGGTGCTTTTCAGGTAC |
| F-β-actin2 | GCAGAAGGAGATCACATCCCTGGC |
| R-β-actin2 | CATTGCCGTCACCTTCACCGTTC |
| F-kcnk5b | ATCACTCTCCTCGTCTGCAACG |
| R-kcnk5b | GAGTCCCATGCACAACGTGCAG |
| F-GFP | AAGGGCATCGACTTCAAGG |
| R-GFP | TGCTTGTCGGCCATGATATAG |

**Table 3.** qRT-PCR primers for human genes.

| F-lef1 | CTACCCATCCTCACTGTCAGTC |
|---|---|
| R-lef1 | GGATGTTCCTGTTTGACCTGAGG |
| F-shh | CCGAGCGATTTAAGGAACTCACC |
| R-shh | AGCGTTCAACTTGTCCTTACACC |
| F-aldh1a2 | GAGTAACTCTGGAACTTGGAGGC |
| R-aldh1a2 | ATGGACTCCTCCACGAAGATGC |
| F-pea3 | AGGAACAGACGGACTTCGCCTA |
| R-pea3 | CTGGGAATGGTCGCAGAGGTTT |
| F-msx1 | GACTCCTCAAGCTGCCAGAAGA |
| R-msx1 | ACGGTTCGTCTTGTGTTTGCGG |
| F-GAPDH | GTCTCCTCTGACTTCAACAGCG |
| R-GAPDH | ACCACCCTGTTGCTGTAGCCAA |

**Table 4.** Primer sequences for mouse qPCR.

| F-lef1 | ACTGTCAGGCGACACTTCCATG |
| --- | --- |
| R-lef1 | GTGCTCCTGTTTGACCTGAGGT |
| F-shh | GGATGAGGAAAACACGGGAGCA |
| R-shh | TCATCCCAGCCCTCGGTCACT |
| F-aldh1a2 | CACAAGACACGAGCCCATTGGA |
| R-aldh1a2 | GGTTTGATGACCACGGTGTTACC |
| F-pea3 | CACAGACTTCGCCTACGACTCA |
| R-pea3 | GCAGACATCATCTGGGAATGGTC |
| F-msx1 | AGGACTCCTCAAGCTGCCAGAA |
| R-msx1 | CGGTTGGTCTTGTGCTTGCGTA |
| F-GAPDH | CATCACTGCCACCCAGAAGACTG |
| R-GAPDH | ATGCCAGTGAGCTTCCCGTTCAG |

procedure is: Step 1: 98℃ for 2 min. Step 2: 98℃ for 10 s, 55℃ for 30 s, and 68℃ for 30 s for 30 cycles. Step 3: 68℃ for 10 min, 4℃ to hold. Then gel electrophoresis was conducted at 120 v for 30 min.

## FRET-FLIM detection and analysis

Hek293T cells were transfected with 1 μg of pcDNA-*kcnk5b*-GFP and 1 μg of the pcDNA6-Kirin-FRET sensor (*Shen et al., 2019*). Fluorescence lifetime imaging measurements were made by photon counting the fluorescence emission of CFP using a two-photon-confocal Hyperscope (Scientifica, UK) and PMT-hybrid 40 MOD five photon detectors (Picoquant, Germany). Counted photon emissions were calculated and analyzed using SymPhoTime 64, version 2.4 (Picoquant, Germany, RRID: SCR_016263).

## Electrophysiology

Transfected HEK293T cells were seeded on glass coverslips (Fisher Brand), and incubated in cell culture medium at 37℃, 5% $CO_2$, 95% relative humidity for 4–6 hr. The seeded coverslips were transferred into Tyrode's solution (138 mM NaCl, 4 mM KCl, 2 mM $CaCl_2$,1 mM $MgCl_2$, 0.33 mM $NaH_2PO_4$, 10 mM Glucose, 10 mM HEPES). Cells were assessed in the ruptured-patch whole-cell configuration of the patch-clamp technique using and EPC9 or EPC10 amplifier (HEKA) with borosilicate glass pipettes (Sutter Instruments) with 3–6 MΩ resistance when filled with pipette solution (130 mM glutamic acid, 10 mM KCl, 4 mM MgCl2, 10 mM HEPES, 2 mM ATP, pH to 7.2). After gigaseal formation, cells were voltage-clamped at −80 mV. Potassium conductance was elicited by test pulses from −100 mV to 70 mV (in 10 mV increments) of 600 ms duration at a cycle length of 10 s. The resulting tracings were converted into itx files by the ABF Software (ABF Software, Inc, RRID: SCR_019222) and then analyzed using Clampfit Software (Molecular Devices, RRID: SCR_011323). Currents were measured at the end of the test pulses.

## Acknowledgements

This work was supported by funding from ShanghaiTech University, and from the Deutsche Forschungsgemeinschaft Grant AN 797/4–1 (CLA). We thank the Microscope Facility of the School of Life Sciences and Technology at Shanghaitech University for their expertise and support. We also wish to thank C Bökel for scientific discussions.

## Additional information

### Funding

| Funder | Grant reference number | Author |
|---|---|---|
| Deutsche Forschungsge-meinschaft | AN 797/4-1 | Christopher L Antos |
| ShanghaiTech University | | Christopher L Antos |

The funders had no role in study design, data collection and interpretation, or the decision to submit the work for publication.

### Author contributions

Chao Yi, Data curation, Formal analysis, Methodology, Writing - review and editing; Tim WGM Spitters, Data curation, Investigation, Writing - review and editing; Ezz Al-Din Ahmed Al-Far, Formal analysis, Investigation, Writing - review and editing; Sen Wang, TianLong Xiong, Simian Cai, Xin Yan, Investigation; Kaomei Guan, Ali El-Armouche, Resources, Writing - review and editing; Michael Wagner, Formal analysis, Investigation, Methodology, Writing - review and editing; Christopher L Antos, Conceptualization, Resources, Data curation, Formal analysis, Supervision, Funding acquisition, Investigation, Methodology, Writing - original draft, Project administration, Writing - review and editing

### Author ORCIDs

Christopher L Antos (ID) https://orcid.org/0000-0001-8881-8568

### Ethics

Animal experimentation: This study was performed in strict accordance with guidelines for the care and use of laboratory animals for the European Union, Germany, Landesdirektion Sachsen, the Technische Universität Dresden, China and ShanghaiTech University. The protocols were approved by the Landesdirektion Sachsen (Permit number: DD24.1-5131/394/79) and the Shanghaitech Ethical Use of Aminals Committee (20200903003) All procedures using zebrafish were performed under Tricane anesthesia, and every effort was made to minimize discomfort and suffering.

### Decision letter and Author response

Decision letter https://doi.org/10.7554/eLife.60691.sa1
Author response https://doi.org/10.7554/eLife.60691.sa2

## Additional files

### Supplementary files

• Transparent reporting form

### Data availability

All data generated are included in the manscript and files.

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
