## [Decision Letter]

**Acceptance summary:**

The manuscript investigates the relationship between calcineurin and Kcnk5b in regulation of allometric versus isometric fin growth in zebrafish. The manuscript also presents interesting results supporting a cell-autonomous effect of Kcnk5b, rather than non-autonomous bioelectric action, proposed for channels by several earlier publications in the field. Towards understanding the molecular mechanism via which Kcnk5b activity regulates fin growth the authors provide evidence of changes in Wnt and Shh signaling and/or expression, providing a model for how Kcnk5b could exert its effects. Finally, Ser to Alanine and Ser to Glutamic acid mutations, – nice transgenic and cell culture work, support a model that regulation of Ser435 phosphorylation by Calcineurin regulates fin growth. Therefore, this work advances understanding of how potassium channels regulate allometric fin growth, which would be of interest to the scientific community.

**Decision letter after peer review:**

Thank you for submitting your article "An endogenous scaling mechanism that controls a K^+^-leak channel to regulate morphogen and growth factor transcription" for consideration by *eLife*. Your article has been reviewed by 3 peer reviewers, including Lilianna Solnica-Krezel as the Reviewing Editor and Reviewer #1, and the evaluation has been overseen by Marianne Bronner as the Senior Editor.

The reviewers have discussed the reviews with one another and the Reviewing Editor has drafted this decision to help you prepare a revised submission.

Summary:

Mechanisms regulating proportional growth during development and regeneration are poorly understood. The manuscript investigates the relationship between calcineurin and Kcnk5b in regulation of allometric versus isometric fin growth in zebrafish. The manuscript also presents interesting results supporting a cell-autonomous effect of Kcnk5b, rather than non-autonomous bioelectric action, currently prevailing model. Towards understanding the molecular mechanism via which Kcnk5b activity regulates fin growth the authors provide evidence of changes in Wnt and Shh signaling and/or expression, providing a model for how Kcnk5b could exert its effects. Finally, Ser to Alanine and Ser to Glutamic acid mutations in transgenic zebrafish and cell culture, support a model that regulation of Ser435 phosphorylation by Calcineurin regulates fin growth. This work advances understanding of how potassium channels regulate allometric fin growth.

Essential revisions:

The manuscript reports large amount of data, which is in general well presented. However, several conclusions require further experimental support, toning down and/or clarification. These and other concerns need to be addressed before the manuscript becomes suitable for publication.

1. In the manuscript, the control for the heat-shock Tg[hsp70:kcnk5b-GFP] transgenic animals is the transgenic line not subjected to heat shock. Since heat-shock is known to induce significant transcriptional changes, the authors need to repeat all experiments comparing heat-shocked wild-types with heat-shocked transgenics. This is particularly important since some of the results are unexpected (msx being up after 6, but not 12 hours, aldh1 strongly up after 6 but down after 12 hours).

2. The gene expression changes presented in Figure 1 are impressive. However, interpreting the changes in lef1 as evidence for alteration of Wnt signaling, and changes of mxb as alterations of BMP signaling, requires further support. While lef1 is a good direct Wnt target in many tissues, it can be regulated in the fin in a Wnt independent manner (Wehner et al. Cell Reports 2014). Moreover, it is not clear that msxb is only regulated by BMP signaling and thus could serve as readout for pathway activity. Similar to the larval experiments, authors should use a Wnt reporter to support their findings that Wnt signaling is regulated. For Hedgehog signaling, patched is often used as readout, and authors should also assay for its expression.

3. The authors conclude increased nuclear localization of b-catenin in Tg[hsp70:kcnk5b-GFP] upon HS, without altered overall b-catenin fluorescence intensity. However, in the presented panels anti-β catenin intensity appears higher in the HS transgenic tissue. Figure 1H: instead of showing single channels for the overview, single channels should be shown for the higher mag, so readers can assess themselves whether there is nuclear β-catenin. Also, please clarify the thickness of sections and whether we are looking at a regular IFF or confocal sections?

4. The mosaic experiments are interesting, but their description and documentation should be improved. Moreover, the conclusions are not fully supported by the data.

– Line 193-194 The authors state "Moreover, in all tissues, the ectopic mCherry expression was always limited to the Kcnk5b-positive cells (Figure 3B-D). However, the graph in Figure 3D, show many more mCherry positive cells than GFP positive cells? Isn't this consistent with the non-autonomous model? 10)

– Second, why are the numbers of mCherry+ cells much higher in 3 of the counted tissues in the graph in D than of the GFP+ cells? If this included endogenous mCherry expression, including endogenous expression domains is meaningless for the purpose of showing whether induction is cell-autonomous. Authors must concentrate on domains where there is no endogenous mCherry "background".

– Third, it needs to be considered that Wnt ligands induced by transgene expression might not spread far considering their hydrophobic nature. Thus, proving that there is no mCherry induction outside the GFP domain must be done at very high cellular resolution and not in those low-resolution images the authors provide.

– Figure 3A cartoon needs to be revised as the text states that "we raised mosaic embryos as larva (should be larvae), but the figure shows analyses of mosaic blastulae. When exactly at the larval stage heat shock has been applied? On line 185 the authors talk about "chimeric embryos" – please clarify. Also, please provide the number of experimental "chimeric animals (embryo or larvae)".

5. It's interesting that authors can detect increased growth of the fin fold relative to the body, although the channel and the presumed downstream signals are induced everywhere. So what about overall larval growth? Does this also increase?

6. Based on the results presented in Figure 4, the authors conclude that "Transfection of HEK cells with these two other channels resulted in the same transcriptional profile as Kcnk5b. However, similar changes were observed for SHH and PEA3, but not for LEF1, ALDH1a2 or MSX1, contrasting the results for kcn5b in HEK293T in Figure 4Bb. This undermines the authors' conclusion that "that the transcriptional response to Kcnk5b is a response to the electrophysiological changes associated with intracellular K^+^ 221 leak." To further test this, such comparative experiments should be performed also in HeLa and N2 cells, in which Kcn5b channel induces distinct transcriptional responses.

7. Authors assay for gene expression in stably transfected cells. Continuous expression of the channel might have many effects on the physiology / proliferation of the cells, which preclude meaningful conclusions from transcriptional changes. Experiments should also be performed with inducible expression systems.

8. The experiments testing the specificity of Calcineurin regulation of Kcnk5b are compelling as are the experiments testing the role of S345 and its phosphorylation. However, Ser345 phosphorylation can be only inferred from these experiments. To make this conclusion, a more direct evidence of S345 would be required. Therefore, it is an overstatement "we observed a linear relationship between the proportional length of the unamputated lobes and the phosphorylation status of the channels", this should be revised as "presumptive phosphorylation".

9. According to Figure 6B the difference between the proportional length of the unamputated lobes for wild-type Kcnk5b channel and the phosphomimetic S345E form is not significant. This could be due to relatively small number of experimental animals; however, the authors need to acknowledge this. It is also possible that in vivo effect of this mutation are not as strong as suggested by electrophysiology. The most elegant way to test effects of these mutations would be to edit them into the endogenous kcnk5b locus, although this would require significant amount of time and is not essential.

10. The authors conclude "kcnk5bS345E mutant was resistant to calcineurin-mediated inhibition (Figure 6D)". However, it appears that Calcineurin addition enhanced the mutant channel activity? Are these differences significant?

11. The number of experimental animals, and the number of experiments performed should be clearly indicated in figures or figure legends.

[Editors' note: further revisions were suggested prior to acceptance, as described below.]

Thank you for submitting your article "An endogenous scaling mechanism that controls a K^+^-leak channel to regulate morphogen and growth factor transcription" for consideration by *eLife*. Your article has been reviewed by 2 peer reviewers, and the evaluation has been overseen by a Reviewing Editor and Marianne Bronner as the Senior Editor.

The reviewers have discussed the reviews with one another and the Reviewing Editor has drafted this decision to help you prepare a revised submission.

We would like to draw your attention to changes in our policy on revisions we have made in response to COVID-19 (https://elifesciences.org/articles/57162). Specifically, when editors judge that a submitted work as a whole belongs in *eLife* but that some conclusions require a modest amount of additional new data, as they do with your paper, we are asking that the manuscript be revised to either limit claims to those supported by data in hand, or to explicitly state that the relevant conclusions require additional supporting data.

Essential revisions:

There was consensus among the reviewers that the revised manuscript was significantly improved. However, some conclusions in the Abstract and/or the main manuscript require tempering down and/or clarification. The following concerns need to be addressed before the manuscript becomes suitable for publication.

While the authors have now added the appropriate controls at least for some experiments, they do not adjust their conclusions based on the new data. While an upregulation is observed of lef1 and shh after 6 hours, albeit much less dramatically as when compared to non-heat shocked fish (about 1.5 to 2 fold with the appropriate new controls, compared to 30 to 40 fold in the old data), the effect after 12 hours has disappeared. In fact, some of the results like downregulation of lef1 after 12 hours have disappeared as well. However, the authors still state "The elevated expression of shh and lef1 continued 12 hours after the single pulse (Figure 1Ba Figure 1—figure supplement Figure 1B)". This is clearly not what the Figure 1—figure supplement Figure 1B.

The new results representing properly-controlled experiments should be moved to the main figures. These new results need to be appropriately incorporated into the authors conclusions.

Appropriate controls using heat-shocked WT without the Tg[hsp70:kcnk5b-GFP] transgene, should be also performed for other experiments, e.g. much of figure 2.

In their response letter, the authors agree with the reviewers that "lef1 activation may be independent" of Wnt signaling and alone cannot be used as a reporter for Wnt signaling. The authors changed some of the conclusions in the Results. However, the abstract still states "We show the activation of Kcnk5b is sufficient to activate several development programs, the most responsive of which are shh and lef1". "Developmental programs" is an overstatement and this needs to be rephrased. Similarly, the Abstract states "Kcnk5b can induce the expression of these developmental programs in cultured mammalian cell lines". The phrase "developmental programs" should be removed as it implies a gene or signaling cascade(s) for which there is insufficient evidence.

In response to the request to clarify whether data points in qPCR graphs are biological replicates, the authors claim they added that information, but it is very unclear. E.g in the legend to Figure 1 they write "…each experiment represents an N of 3 or more, with each N having 2 or more replicates". What does that mean? One is concerned that "replicates" are technical replicates and the impressively large sample sizes used for the qPCR experiments throughout the paper include technical replicates. In the response letter the authors state that sufficient number of biological replicates have been performed. This needs to be clearly presented in the manuscript.

Moreover, the authors agree with the reviewers that posttranslational modification of serine 345 in Kcnk5b is "presumptive" Yet, the Abstract states "We also demonstrate how post translational modification of serine 345 in Kcnk5b by calcineurin regulates channel activity and controls these developmental programs to scale the fin." The authors discuss their efforts to generate appropriate transgenic or genome edited lines to more directly test this presumptive serine modification. These are challenging experiments. However, without such experiments, one cannot make these definitive conclusions in the Abstract (or elsewhere in the manuscript).

---

## [Author Response]

Essential revisions:The manuscript reports large amount of data, which is in general well presented. However, several conclusions require further experimental support, toning down and/or clarification. These and other concerns need to be addressed before the manuscript becomes suitable for publication.1. In the manuscript, the control for the heat-shock Tg[hsp70:kcnk5b-GFP] transgenic animals is the transgenic line not subjected to heat shock. Since heat-shock is known to induce significant transcriptional changes, the authors need to repeat all experiments comparing heat-shocked wild-types with heat-shocked transgenics. This is particularly important since some of the results are unexpected (msx being up after 6, but not 12 hours, aldh1 strongly up after 6 but down after 12 hours).

We understand the concern, and we added new experiments that compare wild-type (AB background genetic strain) heat-shocked fish with Tg[*hsp70*:*kcnk5b*-GFP] additional heat-shocked fish. These data are included as Figure 1—figure supplement Figure 1, and they show the same gene activation trend.

We posit the differences between the 6 and 12 hour time points as evidence of a close relationship between Kcnk5b activity and the transcriptional activity of these genes. From our qRT-PCR experiments, we see almost all of the heat-shock-induced expression of the channel reduced to near background levels by 12 hours, which correlates with the mentioned reduction in the gene expression between 6 to 12 hours.

2. The gene expression changes presented in Figure 1 are impressive. However, interpreting the changes in lef1 as evidence for alteration of Wnt signaling, and changes of mxb as alterations of BMP signaling, requires further support. While lef1 is a good direct Wnt target in many tissues, it can be regulated in the fin in a Wnt independent manner (Wehner et al. Cell Reports 2014). Moreover, it is not clear that msxb is only regulated by BMP signaling and thus could serve as readout for pathway activity. Similar to the larval experiments, authors should use a Wnt reporter to support their findings that Wnt signaling is regulated. For Hedgehog signaling, patched is often used as readout, and authors should also assay for its expression.

As requested, we assess *patched1*, *patched2* and *bmp2* and observed up-regulation (Figure 1—figure supplement Figure 1H). We kept the reference of *msxb* to BMP signaling, because the increased *bmp* transcription that we did see although was slight, it was statistically significant. To address the reviewer’s point about β-catenin regulated transcription, we also examined the expression of other β-catenin-activated genes: *axin2, cyclin D and myc*, but we did not see upregulation (data placed in Suppl. Figure 1I). Thus, as the reviewer suggested might be the case, *lef1* activation may be independent. These results altered our interpretation, and consequently, we have modified our text to focus on *lef1* transcription and less on direct β-catenin-dependent Wnt activation. We still do see more β-catenin nuclear localization from the counted confocal images of the immunohistochemistry sections, and ultimately one can expect β-catenin-dependent Wnt to be activated indirectly, since the phenotype involves patterned fin growth even for the unamputated fins. We provide the following hypotheses to these observations in the Discussion section:

“While we observed that Kcnk5b is sufficient to promote *lef1*-mediated transcription (Figures 1-3), we did not observe direct activation of down-stream target genes of β-catenin activity (Figure 1—figure supplement Figure 1I). We posit that Kcnk5b activity does not directly activate canonical Wnt, but primes cells for increased β-catenin activity upon the reception of a Wnt signal through up-regulation of Lef1. Ultimately, continued Kcnk5b activity must lead to increased β-catenin-dependent Wnt signaling, since evidence indicates that it is required for fin outgrowth (Kawakami et al., 2006; Stoick-Cooper et al., 2007), which is the phenotype we get by Kcnk5b activation in amputated and unamputated fin (Figure E-I).”

3. The authors conclude increased nuclear localization of b-catenin in Tg[hsp70:kcnk5b-GFP] upon HS, without altered overall β-catenin fluorescence intensity. However, in the presented panels anti-β catenin intensity appears higher in the HS transgenic tissue. Figure 1H: instead of showing single channels for the overview, single channels should be shown for the higher mag, so readers can assess themselves whether there is nuclear β-catenin. Also, please clarify the thickness of sections and whether we are looking at a regular IFF or confocal sections?

We understand the reviewer’s concern. The sections are of 10 µm thick and the images are taken by confocal microscopy. Each confocal image as shown in Figure 1H is 0.45 μm. We now include this information in the figure legend: “Confocal planes (0.45 μm) of immunohistochemistry stained 10 μm sections of…” (lines 569-570). We decided to use the sections with the highest intensities and clearest histology for each group for the panels, but this clearly raised a discrepancy in difference in expression. We have gone through all of our stainings and now provide panels that are representative of our general observations and of our quantitative analyses. In addition, we added confocal serial sections as supplemental data (Figure 1—figure supplement Figure 2E-J) and high-light a few nuclei to show what we mean by nuclear β-catenin nuclear staining. We also provided the higher magnification image panels. Each point in the graph represents a combined total of β-catenin nuclear distribution for one confocal section. We sectioned > 6 fins (3 different experiments) for each group, so our quantitative analyses should reliably reflect what is occurring in fin.

4. The mosaic experiments are interesting, but their description and documentation should be improved. Moreover, the conclusions are not fully supported by the data.– Line 193-194 The authors state "Moreover, in all tissues, the ectopic mCherry expression was always limited to the Kcnk5b-positive cells (Figure 3B-D). However, the graph in Figure 3D, show many more mCherry positive cells than GFP positive cells? Isn't this consistent with the non-autonomous model?– Second, why are the numbers of mCherry+ cells much higher in 3 of the counted tissues in the graph in D than of the GFP+ cells? If this included endogenous mCherry expression, including endogenous expression domains is meaningless for the purpose of showing whether induction is cell-autonomous. Authors must concentrate on domains where there is no endogenous mCherry "background".

To clarify the point about ectopic versus normal expression of the mCherry reporter, the graphed mCherry positive cells lacking GFP are endogenous cells that normally expressed the reporter during embryonic/larval development. We included these cells to represent the cells normally expressing β-catenin-Wnt signaling, because we were concerned that if we didn’t, then an alternative argument could be made that there should be some cells that endogenously express the reporter, and there might be concern that the recipient fish were not transgenic for the reporter. However, we do see the review’s point. To address both issues better, we removed the measurements of the endogenous cells and reconfirmed the mCherry genotypes by qPCR on the fish parents and recipient fish harboring the donor cells. All the animals are homozygous for the mCherry reporter.

We are very certain about our conclusion that all ectopic mCherry expression is limited to only the cells expressing Kcnk5b-GFP. Even muscle, which does already express the reporter, shows that the muscle cells expressing Kcnk5b-GFP have elevated cell-autonomous expression of the Lef1-regulated mCherry compare to their neighbors lacking it (Figure 3Bj-l; 3Cr-t). We have provided more evidence from more transplantations as well as higher magnifications (Figure 3Da-c; Figure 3—figure supplement Figure 1Ia-Li) and three-dimensional composites (different axes rotations) from confocal z-stacks (Figure 3Dd-f; Figure 3—figure supplement Figure 1Gh,i-Jh,i) as additional evidence.

– Third, it needs to be considered that Wnt ligands induced by transgene expression might not spread far considering their hydrophobic nature. Thus, proving that there is no mCherry induction outside the GFP domain must be done at very high cellular resolution and not in those low-resolution images the authors provide.

We have provided higher resolution images of individual cells from several different examples and three-dimensional composites from z-stacks (Figure 3D; Figure 3—figure supplement Figure 1Ia-Li) to show that only cells harboring Kcnk5b-GFP induced mCherry expression.

– Figure 3A cartoon needs to be revised as the text states that "we raised mosaic embryos as larva (should be larvae), but the figure shows analyses of mosaic blastulae. When exactly at the larval stage heat shock has been applied? On line 185 the authors talk about "chimeric embryos" – please clarify. Also, please provide the number of experimental "chimeric animals (embryo or larvae)".

We have revised the figure cartoon to show make clear when the heat shocks were applied and that the cells were assessed in 48- and 72-hour old fish. By chimeric animals, we mean the recipient fish embryos that harbored the incorporated the transplanted donor cells. We specified this more clearly in the text. “we observed ectopic activation of 7XTCF-Xla.sam:mCherry reporter only in donor cells of chimeric 48 hpf and 72 hpf fish (recipient 7XTCF-Xla.sam:mCherry fish harboring transplanted cells from Tg[hsp70:*kcnk5b*-GFP;7XTCF-Xla.sam:mCherry] embryos).”

5. It's interesting that authors can detect increased growth of the fin fold relative to the body, although the channel and the presumed downstream signals are induced everywhere. So what about overall larval growth? Does this also increase?

To address this point, we provide body (nose-to-base of somatic musculature) measurements with our dorsal-to-ventral measurements. Unexpectedly, we observe that the body lengths are reduced by the *kcnk5b* expression. Despite this, the average lengths of the anterior-to-posterior finfolds are increased. We include these data in Figure 3—figure supplement Figure 1G and 2H. We speculate that the reduced body size stems from negative effects on different body systems such as the central nervous system and circulatory system. We had to modify the heat-shock method to perform a daily heat-shock protocol, since continued heat shock after the heart formation lead to significant mortality.

6. Based on the results presented in Figure 4, the authors conclude that "Transfection of HEK cells with these two other channels resulted in the same transcriptional profile as Kcnk5b. However, similar changes were observed for SHH and PEA3, but not for LEF1, ALDH1a2 or MSX1, contrasting the results for kcn5b in HEK293T in Figure 4Bb. This undermines the authors' conclusion that "that the transcriptional response to Kcnk5b is a response to the electrophysiological changes associated with intracellular K^+^ 221 leak." To further test this, such comparative experiments should be performed also in HeLa and N2 cells, in which Kcn5b channel induces distinct transcriptional responses.

We understand this point, and we modified our conclusions about the effects of Kcnk5b.

“However, when we either stably expressed or transient transfected Kcnk5b in different cell lines, we did not observe consistent activation of Lef1 or Shh (Figure 4). […] It remains to be explored whether intermolecular interactions between the channels and another protein contribute to the scaling of other organs as well as how other electrophysiological mechanisms that control membrane potential have the same growth effect”

We also did the requested transfection experiments for Kcnk9 and Kcnk10 in Hela and N2A cells, and we observed several profile differences between how Kcnk5b affects transcription and how Kcnk9 and 10 affect transcription from expression analyses. These data support the reviewer’s point. Consequently, we have modified the text in the Results section:

“When we tested whether Kcnk9 and Kcnk10 produce similar transcription profiles as Kcnk5b in HELA and N2A cells, we observed similar profiles between Kcnk9 and Kcnk10 (Figure 4H-L), but differences between their outcomes and the transcriptional outcome of Kcnk5b. […] We propose that the variability in genes that are transcribed may explain why the solitary change in the activity of this channel in all cells of the fin leads to the variable transcriptional responses that are needed to promote coordinated growth of a multi-tissue anatomical structure.”

And in the Discussion section:

“While we observed similarities between Kcnk5b and Kcnk9 or Kcnk10 in HEK cells, in Hela and N2A cells, we observed only partially similar profiles for the selected genes (while Kcnk9 and Kcnk10 were similar) (Figure 4). We postulate that the observed differences may come from one or more of the following explanations. First, different responses from channels of the same ion-type are due to different levels of membrane potential changes, which results in different levels of gene transcription. Second, these channels have different intracellular sequences, which may determine other unknown intermolecular interactions that have different signal transduction properties. In any case, we did observe that transgenic overexpression of Kcnk9 produces a similar allometric growth of the caudal fin (Figure 6K,L)”

7. Authors assay for gene expression in stably transfected cells. Continuous expression of the channel might have many effects on the physiology / proliferation of the cells, which preclude meaningful conclusions from transcriptional changes. Experiments should also be performed with inducible expression systems.

We added transient transfection experiments for Kcnk5b and they show the same responses, albeit, the expression levels are not as high. We included this data in Figure 4—figure supplement Figure 2A,B and referenced it in the text (*seelines* 227-228).

8. The experiments testing the specificity of Calcineurin regulation of Kcnk5b are compelling as are the experiments testing the role of S345 and its phosphorylation. However, Ser345 phosphorylation can be only inferred from these experiments. To make this conclusion, a more direct evidence of S345 would be required. Therefore, it is an overstatement "we observed a linear relationship between the proportional length of the unamputated lobes and the phosphorylation status of the channels", this should be revised as "presumptive phosphorylation".

We understand the reviewer’s point and revised the text to state “presumptive phosphorylation”.

9. According to Figure 6B the difference between the proportional length of the unamputated lobes for wild-type Kcnk5b channel and the phosphomimetic S345E form is not significant. This could be due to relatively small number of experimental animals; however, the authors need to acknowledge this. It is also possible that in vivo effect of this mutation are not as strong as suggested by electrophysiology. The most elegant way to test effects of these mutations would be to edit them into the endogenous kcnk5b locus, although this would require significant amount of time and is not essential.

We agree with this point, and we have increased the numbers to see whether the difference becomes statistically significant. By increasing the numbers, we do reach a statistical significance (Figure 6I). We posit that the lack of growth difference in the unamputated fin is due to the relatively low growth rate of uninjured fins.

We have tried to mutate the endogenous locus a few times using different CRISPR-based methods, but we as of yet have not been able to mutate the proline (serine not possible because the codon does not contain the required specific nucleotides for base switching to alanine/glutamic acid) or to induce recombination with a homologous sequences that contain the desired mutations. While we agree that this is experiment would be a good test, because of continued inability to target the endogenous site, we are thankful that this is not required.

10. The authors conclude "kcnk5bS345E mutant was resistant to calcineurin-mediated inhibition (Figure 6D)". However, it appears that Calcineurin addition enhanced the mutant channel activity? Are these differences significant?

The observed trends are not significant. We incorporated the results of the significance tests to show that apparent differences between any two groups are not significant.

11. The number of experimental animals, and the number of experiments performed should be clearly indicated in figures or figure legends.

We have rechecked each figure legend and placed this information in the legend. We have also provided the raw data values for each experiments in the accompanying files that *eLife* links to the figures.

[Editors' note: further revisions were suggested prior to acceptance, as described below.]

Essential revisions:There was consensus among the reviewers that the revised manuscript was significantly improved. However, some conclusions in the Abstract and/or the main manuscript require tempering down and/or clarification. The following concerns need to be addressed before the manuscript becomes suitable for publication.While the authors have now added the appropriate controls at least for some experiments, they do not adjust their conclusions based on the new data. While an upregulation is observed of lef1 and shh after 6 hours, albeit much less dramatically as when compared to non-heat shocked fish (about 1.5 to 2 fold with the appropriate new controls, compared to 30 to 40 fold in the old data), the effect after 12 hours has disappeared. In fact, some of the results like downregulation of lef1 after 12 hours have disappeared as well. However, the authors still state "The elevated expression of shh and lef1 continued 12 hours after the single pulse (Figure 1Ba Figure 1—figure supplement Figure 1B)". This is clearly not what the Figure 1—figure supplement Figure 1B.The new results representing properly-controlled experiments should be moved to the main figures. These new results need to be appropriately incorporated into the authors conclusions.

We understand the concerns and apologize for the misunderstanding. Since previously kept the original results in the main figure 1, so we maintained the original text. We have now replaced these figure panels with the previous supplementary panels as requested and have revised the text accordingly (lines 129-149). We have also replaced the mRNA in situ experiments for *shh* and *lef1* (Figure 1E and F) with experiments that contain the heat-shock wild-type controls to make clear that heat shock alone does not induce the expression of these genes. The expression patterns are the same as the previous data (which showed that transgenic fish that were not heat shocked do not induce the expression of these genes).

Appropriate controls using heat-shocked WT without the Tg[hsp70:kcnk5b-GFP] transgene, should be also performed for other experiments, e.g. much of figure 2.

We added the requested controls to those figure panels that lacked them and joined them in one figure panel (Figure 2A). We have also provided new in situ results for *shh* expression the fish pectoral fin bud (Figure 2B) to further support the qRT-PCR results (now with the additional heat shock controls). The other figure panels already included non-transgenic heat-shocked fish.

In their response letter, the authors agree with the reviewers that "lef1 activation may be independent" of Wnt signaling and alone cannot be used as a reporter for Wnt signaling. The authors changed some of the conclusions in the Results. However, the abstract still states "We show the activation of Kcnk5b is sufficient to activate several development programs, the most responsive of which are shh and lef1". "Developmental programs" is an overstatement and this needs to be rephrased. Similarly, the Abstract states "Kcnk5b can induce the expression of these developmental programs in cultured mammalian cell lines". The phrase "developmental programs" should be removed as it implies a gene or signaling cascade(s) for which there is insufficient evidence.

We used the “developmental program” term, not only because of the activation of the genes important for development, but that ultimately the all developmental programs required for fin growth (which the formation/development of new tissues) must be activated to induce the allometric growth phenotype that we observe in Figure 6. However, we understand the concerns, so we made the requested changes and hope that our current conclusions address the issue.

In response to the request to clarify whether data points in qPCR graphs are biological replicates, the authors claim they added that information, but it is very unclear. E.g in the legend to Figure 1 they write "…each experiment represents an N of 3 or more, with each N having 2 or more replicates". What does that mean? One is concerned that "replicates" are technical replicates and the impressively large sample sizes used for the qPCR experiments throughout the paper include technical replicates. In the response letter the authors state that sufficient number of biological replicates have been performed. This needs to be clearly presented in the manuscript.

We apologize for the ambiguity. We detailed the representation of our data.

Moreover, the authors agree with the reviewers that posttranslational modification of serine 345 in Kcnk5b is "presumptive" Yet, the Abstract states "We also demonstrate how post translational modification of serine 345 in Kcnk5b by calcineurin regulates channel activity and controls these developmental programs to scale the fin." The authors discuss their efforts to generate appropriate transgenic or genome edited lines to more directly test this presumptive serine modification. These are challenging experiments. However, without such experiments, one cannot make these definitive conclusions in the Abstract (or elsewhere in the manuscript).

We apologize for missing this change. We have revised accordingly.